# Broadband Circular Polarised Printed Antennas for Indoor Wireless Communication Systems: A Comprehensive Review

**DOI:** 10.3390/mi13071048

**Published:** 2022-06-30

**Authors:** Ahmed Jamal Abdullah Al-Gburi, Zahriladha Zakaria, Hussein Alsariera, Muhammad Firdaus Akbar, Imran Mohd Ibrahim, Khalid Subhi Ahmad, Sarosh Ahmad, Samir Salem Al-Bawri

**Affiliations:** 1Center for Telecommunication Research and Innovation (CeTRI), Faculty of Electronics and Computer Engineering (FKEKK), Universiti Teknikal Malaysia Melaka (UTeM), Durian Tungal 76100, Malacca, Malaysia; ahmedjamal@utem.edu.my (A.J.A.A.-G.); imranibrahim@utem.edu.my (I.M.I.); 2School of Electrical and Electronic Engineering, Universiti Sains Malaysia, Nibong Tebal 14300, Penang, Malaysia; firdaus.akbar@usm.my; 3Department of Electronic Engineering, Northern Technical University (NTU), Mosul 41002, Iraq; jarkovo_1988@ntu.edu.iq; 4Department of Signal Theory and Communications, Universidad Carlos III de Madrid, 28911 Leganes, Spain; saroshahmad@ieee.org; 5Space Science Centre, Climate Change Institute, Universiti Kebangsaan Malaysia (UKM), Bangi 43600, Selangor, Malaysia; samir@ukm.edu.my

**Keywords:** printed antennas, circularly polarised (CP), indoor antennas, linearly polarised (LP), indoor wireless communication (IWC), phase difference (PD)

## Abstract

With the rapid changes in wireless communication systems, indoor wireless communication (IWC) technology has undergone tremendous development. Antennas are crucial components of IWC systems that transmit and receive signals within indoor environments. Thus, the development of indoor technology is highly dependent on the development of indoor antennas. However, indoor environments with limited space require the fewest indoor antenna units and the smallest indoor antenna sizes possible. Hence, indoor antennas with compact size and broad applications have become widely preferred. In an IWC system, circularly polarised (CP) antennas are generally important, especially in dense indoor environments, because compared with linearly polarised (LP) antennas, CP antennas reduce polarisation mismatch and multipath losses. This paper combs through the existing studies related to three-dimensional (3D) geometry (nonplanar) or waveguide indoor antennas and the two common approaches to two-dimensional (2D) geometry (planar) indoor antennas, namely, broadband CP printed monopole antennas (BCPPMAs) and broadband CP printed slot antennas (BCPPSAs). The advantages, disadvantages and limitations of previous works are highlighted as well. These research works are summarised, compared and analysed to understand the recent specifications of BCPPMAs and BCPPSAs to generate the most appropriate design structure suitable for current IWC systems.

## 1. Introduction

Wireless communication technologies (WCTs) are usually utilised within buildings and are known as indoor wireless technologies. Distributed indoor antennas are important components of indoor wireless communication (IWC) systems. Figure 1, which was redrawn from [1], shows an indoor distributed antenna system (IDAS) consisting of a headend and an active indoor network. The headend receives radio frequency (RF) signals from one or more base transceiver stations and combines them into a single RF signal. The active network comprises the RF/optical converter, optical/digital converter and remote unit (RU). The RF/optical converter converts RF signals into optical signals, splits them into many optical signals and then sends them to the optical/digital converter through fibre cables. The optical signals, once received by the optical/digital converter, are converted to digital signals and sent to the RU in the active network through fibre cables. The RU comprises a digital/RF converter and RF power amplifier. The digital signals are converted to RF signals and then amplified via the RU. Indoor distributed antennas are installed near RUs and are connected to them through short RF jumper cables (around 1 m) [1].

The employment of distributed indoor antennas inside buildings is rapidly growing due to the active development of wireless communication technologies, such as GSM, 3G, LTE, 5G and WLAN. With the equally rapid increase in the numbers of wireless terminals inside buildings, airports and conference rooms, the use of distributed indoor antennas has broadened. However, this development has made IDASs increasingly complex and expensive. Hence, the number of elements covering various frequency operations should be reduced. Furthermore, researchers are faced with the challenge of designing indoor antennas with suitable polarisation, small size, low profile, omnidirectional radiation pattern and sufficient impedance bandwidth (IBW) [2,3] to cover most wireless communication bands as presented in Table 1.

In wireless communication systems, the positions of user terminals are generally not fixed. They might be randomly oriented to transmitter antennas, thereby weakening the receiving signals; this condition is known as the polarisation mismatch effect [4]. Figure 2 was redrawn from [5] and described the IWC propagation scenario within an indoor environment. The probability of this effect occurring increases in an indoor environment because transmitted signals are reflected from ceilings, walls and other obstacles, as observed in Figure 2. Hence, the original signals that reach the user terminals exhibit different polarisation senses [5,6].

In an indoor environment, radio signals travelling through dense conditions from the transmitter to the receiver regularly experience reflection or refraction caused by obstacles and obstructions, such as walls, furniture and equipment. As a result of reflection or refraction, the transmitted signals travel through different paths to reach the receiver antenna. Hence, the original signals are added with signals with undesirable specifications. Moreover, the signals reach the receiver at different times (see Figure 2). These phenomena affect the quality of the received signals on particular occasions and hinder effective and reliable communication. The effects described herein are multipath fading effects. One solution to the problems of polarisation mismatch and multipath fading involves using indoor antennas with circular polarisation. Circularly polarised (CP) antennas are in-sensitive to equipment orientation [7,8]. Thus, receiver antennas can receive radiation waves from various directions because CP waves have vertical and horizontal electric field components; when these waves are reflected or refracted, the CP senses change. For example, the right-hand circular polarisation (RHCP) of an original signal changes to left-hand circular polarisation (LHCP) [5] and vice versa. A linearly polarised (LP) receiver antenna can receive a CP wave regardless of its orientation [9]. Moreover, circular polarisation could minimise multipath losses [10,11,12].

When used with a certain type of antenna that conforms to indoor environments, such as a conventional waveguide and multilayer planar antennas [13,14,15], they can obtain a wide or broad IBW, omnidirectional radiation pattern, high antenna gains and a wide CP band. However, they suffer from several drawbacks such as complicated design, bulky size, high manufacturing cost and complex design and fabrication. Therefore, alternative antenna structures or techniques are needed to overcome the disadvantages of conventional indoor antennas.

Printed antennas are receiving considerable attention with regard to their use in indoor environments because of their compact size, low profile, simple design, uncomplicated design and fabrication and low production cost [10,16,17,18], in addition to their capabilities of generating CP waves whilst retaining a compact antenna size [19,20] and providing bidirectional radiation patterns, RHCP and LHCP; these properties are essential for antennas which are used in indoor environments [21]. However, these types of antennas suffer from the difficult trade-off between the achieved IBW and axial ratio bandwidth (ARBW) and their size [22,23,24,25,26].

In the last few years, several antenna shapes printed with different techniques have been proposed in response to the increasing demand for compact antennas with wide IBW and ARBW. They include printed slot antennas and printed monopole antennas with biplanar and uniplanar structures. In a biplanar antenna structure, the antenna parts are printed on different sides of the substrate. In a uniplanar antenna structure, the antenna parts are printed on only one side of the substrate; this design is known as coplanar waveguide (CPW) feeding.

The methods for designing printed antennas with a CPW-fed structure represent a vital research area because such an antenna structure possesses several attractive features, such as simplicity, low profile, compact size, good radiation characteristics and low radiation loss. Moreover, using antennas with a CPW-fed structure contributes to the miniaturisation of wireless communication devices because their structural geometry (uniplanar) facilitates their integration with RF/microwave circuits and edge-fed connector boards [27,28]. Table 1 presents various wireless communication bands that are covered by the proposed CP antennas, thus making them suitable candidates for any RF communication system, particularly IWC systems.

**Table 1 micromachines-13-01048-t001:** Frequency bands for several wireless communication systems.

Type of Wireless Communication	Frequency Band (MHz)		Reference
Second-generation systems (2G)	GSM 1800 DCS (1710–1880) GSM 1900 PCS (1850–1990)		[29,30,31]
Third-generation systems (3G)	WCDMA, TD-SCDMA and CDMA 2000 (1880–2170)		[32,33]
Long-term evolution (LTE) systems (4G)	Middle band	LTE 2300 (2300–2400) LTE 2500 (2500–2690)	[33,34]
	Higher band	LTE 3600 (3400–3800)	[35,36]
5G sub-6 GHz	N78 (3300–3800) N79 (4400–5000)		[37,38,39,40]
Wireless local area network (WLAN)	2400 (2400–2500) 5000 (5100–5900)		[41,42]
Worldwide interoperability for microwave access (WiMax)	2300, 2500 and 3500		[35,42,43]

## 2. Overview of Indoor Antennas

Indoor antenna technologies have become widely popular in recent years due to the development of wireless communication technologies, such as WiMAX, 3G, 4G, 5G and WLAN. In addition to the rapid increase in the numbers of wireless terminals inside buildings, airports and conference rooms, researchers are compelled to design antennas that can meet the essential requirements for indoor environments, including suitable polarisation, sufficient IBW to cover most wireless communication bands, omnidirectional radiation patterns with bidirectional radiation and compact size [44]. These requirements have been extensively studied over wideband and broadband antennas. Many antenna topologies, such as monopole and dipole antennas, provide wide or broad IBW. This chapter presents a review of the various techniques that have been employed to develop broadband antennas. It also discusses the advantages and disadvantages of these techniques to understand their recent developments. Solutions for producing new designs that are sufficient for use in modern indoor wireless applications are also investigated.

Conventional indoor antennas, such as indoor waveguide antennas, are mostly nonplanar structures that offer wide IBW with omnidirectional radiation property, high gain and the capability of providing circular polarisation. Several methods and techniques have been used to enhance the IBW and reduce the sizes of LP and CP nonplanar indoor antennas, as presented in Section 2.1 and Section 2.2.

### 2.1. Linearly Polarised Nonplanar Indoor Antennas

K. He [45] presented an indoor antenna structure as shown in Figure 3. The main parts of the proposed design structure include a circular ground plane, truncated cone feeding, three radiator patches connected to the truncated cone and three loaded coupling patches connected to the ground plane by three shorted strips. The ripple levels of the radiation patterns for the horizontal plane are less than 1.5 dB at 1.0, 1.8 and 2.4 GHz. Though the ripple levels increased at the higher frequency, Figure 3c shows the characteristic of omnidirectional radiation in the H-plane.

According to the authors, the circular sleeve (circular ring) is placed on top of the circular ground plane for further IBW enlargement. The overall dimensions of the proposed antenna are 168 mm × 168 mm × 73 mm. The measurement results indicate that this antenna provides a broad IBW ranging from 1 GHz to 7 GHz with an omnidirectional radiation pattern and stable gain of 2.1–7.3 dBi. Another work presented the same method (circular ring) for IBW enlargement [46].

A different method was used by [47] to realise a wide IBW whilst keeping the antenna size compact. As shown in Figure 4, the proposed antenna mainly consists of two substrates which are separated by four capacitive feeds that function by connecting the four-way power divider with four conducting strips and a top shorted strip loading over the circular radiator patch. The overall dimensions of the proposed antenna are 126 mm × 126 mm × 41 mm. The proposed antenna provides a wide IBW of 65.4% ranging from 1.46 GHz to 2.88 GHz, as well as an omnidirectional radiation pattern with an antenna gain of around 5 dBi. The simulated and measured radiation patterns of the proposed design are shown in Figure 4b. It can be clearly seen that the omnidirectional property contributes to the in-phase feed and magnetic field distribution on the circular patch.

For use by [47], the antenna is loaded with a top shorted strip to improve the IBW whilst reducing the overall size. The improvement is due to the coupling between the circular patch and the top shorted strip; it decreases the Q factor of the antenna, thereby resulting in a wide IBW. To broaden the IBW, the authors in [14] added four conducting legs located on a circular disc radiator. They used them to extend the electric length of the antenna, thereby broadening the IBW. In the method proposed by [33,48], parasitic strips are introduced around the antenna radiator, and a tapered balun is used to improve the IBW.

The aforementioned indoor antennas meet most of the recent indoor antenna requirements, such as a wide or broad IBW, omnidirectional radiation pattern and good gain. However, they fail to meet the required size reduction due to their large dimensions and circular polarisation property because the proposed antennas offer vertical or horizontal LP waves.

To meet all the requirements of modern in-building wireless coverage applications, recent works mainly focus on indoor antennas with circular polarisation and wide or broad ARBW whilst maintaining their compact size.

### 2.2. Circularly Polarised Nonplanar Indoor Antennas

The pure polarisations of planar and nonplanar monopole and dipole antennas are linear [10,49,50,51]. Therefore, applicable techniques are needed to design indoor antennas with a circular polarisation property. However, generating CP radiation requires equal amplitude and a 90° PD (phase difference) between two electric field components in the vertical (EVer) and horizontal (EHor) directions.

As presented by Tran and Park in [51], a dipole antenna was designed with a new shape (barbed-shape) and an artificial ground plane to reduce the antenna size and improve the antenna gain and bandwidth. The dual-wideband CP antenna is excited using the crossed bowtie technique, which involves changing the dipole length and the ring diameter, as shown in Figure 5. In sum, the designed dipole antenna can realise wide IBW with dual-band circular polarisation with acceptable ARBWs of 19.3% and 33.8% in the lower and upper bands, respectively. A broadband CP is also obtained with an ARBW of 91% by employing the same technique (crossed bowtie) as that used by Feng in [52].

Quadrilateral dipole patches (kite-shaped patches) and bowtie-shaped feedline were used by [15], as shown in Figure 6, to realise a wide IBW. The method for generating circular polarisation involves two bent slots which are introduced to one kite-shaped patch due to the 90° PD achieved after employing the two slots. The proposed design can realise CP waves with two rotation directions (LHCP and RHCP) simply by introducing impeding slots into the other kite-shaped patch. However, the proposed antenna offers an IBW of 62.6% and ARBW of 37% with an antenna gain of 6 dBi. Other current works on the nonplanar indoor antenna with different techniques and shapes were reported by [53,54,55,56].

Conventional indoor antennas (waveguide, nonplanar or multilayer planar antennas) exhibit a wide or broad IBW, narrow ARBW, high gain and omnidirectional radiation pattern. However, they have a complicated design, bulky antenna size, high manufacturing cost and difficult design and fabrication. Hence, alternative antenna structures or techniques have been proposed to overcome these disadvantages.

With the widespread interest in indoor antenna technology, designing indoor antennas with a planar structure has attracted growing attention. A microstrip or printed antenna offers several advantages, including compact size, low profile, simple design, easy fabrication and low production cost [57,58,59]. In addition to the recent requirements of in-building wireless communication systems, this type of antenna can provide circular polarisation with an omnidirectional radiation pattern [60]. Conversely, traditional microstrip printed antennas suffer from a narrow CP band (the ARBW is limited) and a difficult trade-off between the achieved IBW and the antenna dimensions [60] and between the achieved omnidirectional radiation pattern with two circular polarisation senses (LHCP and RHCP) and the acceptable value of antenna gain [61,62,63,64].

## 3. Microstrip Patch Antenna

The microstrip patch antenna is the most widely used in recent wireless communication systems particularly because of its attractive features, such as compact size, light weight and low cost [65,66,67]. The main contributing factor for microstrip antennas is the comfort and the easy integration with wireless communication systems [68]. In a microstrip printed antenna, the feed line, radiator element and ground plane are printed on a printed circuit board. These antenna parts can be printed on two sides of the PCB (biplanar geometry) or on one side of the PCB (uniplanar geometry). They are etched on a dielectric substrate with known material, dimensions and properties (loss tangent, dielectric constant and permittivity).

Microstrip antennas can be also referred to as printed or patch antennas [69,70,71]. The radiator patch of a printed antenna can be rectangular, square, circular, elliptical, thin strip or any other configuration. Rectangular and circular microstrip patches are the most popular configurations because of the ease of their analysis, design and fabrication. They also have attractive radiation characteristics, especially the radiation with low cross-polarisation. Circular and linear polarisations can be realised with one or more arrays of microstrip antennas [70].

The major drawbacks of microstrip antennas include the difficult trade-off between the achieved bands (IBW and ARBW) and their size, low efficiency and poor polarisation purity antennas [72,73]. Nevertheless, several methods can be used to improve antenna performance and expand the IBW. For example, the length of the antenna substrate can be increased to improve antenna efficiency and widen the IBW. Other methods for generating the CP property of antennas and widening the ARBW include cutting the opposite corners of rectangular or square patches, using elliptical radiator patches, adding stubs on the ground plane and embedding slots into the ground plane [70].

## 4. Broadband CP Printed Antenna (BCPPA)

The use of a single antenna to cover multiple frequency bands is becoming increasingly desirable because such usage helps reduce the cost and complexity of wireless communication systems. Antennas with a broad IBW are desirable for modern communication systems with high data rates, and printed antennas are widely used in modern wireless communications due to their attractive features, such as small size, low profile, easy fabrication and capability of providing a broad IBW [74,75,76].

Modern wireless communication systems require circular polarisation more than they do linear polarisation, especially when the transmitter and receiver antennas are not in polarisation alignment, because a CP antenna can reduce the polarisation mismatch between the transmitter and the receiver [77,78]. Moreover, using CP antennas can minimise the Faraday rotation effect to provide a stable communication link and to mitigate multipath interference [79,80], BCPPAs are preferred in modern wireless communication as they can meet all existing requirements. BCPPAs are thus suitable choices for compact antennas with a broadband CP property and compact patch antennas with broad IBW and ARBW.

Two common approaches are used to design BCPPAs: slot antenna geometry and monopole antenna geometry. These geometries may be biplanar or uniplanar (CPW-fed). In this review, the working principles, the recent techniques for achieving broad IBW and generate CP property and the analysis of measurement results for several recent works are discussed.

### 4.1. Broadband CP Printed Slot Antennas (BCPPSAs)

A slot antenna is a radiating element formed by a slot on a conducting surface. It is designed by cutting an opening on an antenna ground plane. Slot antennas fall into two categories: CPW feeding and microstrip feeding [81,82] (Figure 7 and Figure 8, respectively).

Today, BCPPSAs are popular geometrical configurations which are used for modern wireless communication due to their advantages such as compact size, low profile, broad IBW and ARBW and ease of fabrication and integration [83,84]. This section provides a state-of-the-art review of recent studies on different slot antenna topology designs and techniques that have been used to develop BCPPSAs. The review should help us understand the performance improvements achieved by employing such designs and techniques. The aim of this review is to establish the current research objectives pertaining to slot antenna design and to understand most of the techniques that are used to generate the CP property and broaden the IBW and ARBW. From Figure 7b, it is evident that the proposed antenna model operates as a left-hand circularly polarised antenna (LHCP). Moreover, the radiation pattern is slightly tilted due to the asymmetrical feeding technique that the different gap introduces between the feed line and asymmetric slot.

Given the attractive features of BCPPSAs, many antenna researchers tend to use them to achieve broad IBW and ARBW. Moreover, they carry out different modifications to ordinary slot antenna designs to improve the overall antenna performance and reduce the size of the antenna structure.

Nosrati and Tavassolian [85] proposed a biplanar slot antenna with a U-shaped feed line to produce the first impedance response. The IBW was improved, and circular polarisation was generated by connecting half of a Y-shaped strip to a square slot ground (back side) whilst the remaining half of the Y-shaped strip and a small horizontal stub were printed on the opposite substrate side to be employed as parasitic strips, as shown in Figure 9a,b. These improvements increased the length of the current paths because of the added strip and stub. In addition, the employment of the U-shaped feed line and Y-shaped strip played an essential role in reducing the antenna size. Another U-shaped resonator was also proposed by [86].

Although the proposed antenna provides a large IBW which covers many of the upper-frequency bands of wireless communication systems, such as WLAN and WiMAX, with acceptable gain, its ARBW is narrow; thus, the antenna is unable to provide a broad CP bandwidth for the achieved broad IBW (the overlapping ratio between the two bands is 48%). As observed in Figure 9c, a bidirectional radiation pattern (LHCP and RHCP) was produced with more than 10 dB isolation. However, the radiation patterns were slightly distorted and tilted, possibly because of the use of the asymmetric ground plane.

A novel slot antenna was proposed by [87]. The first IBW of 3.5 GHz was produced using a straight microstrip feed line. A wide rectangular slot was also cut in the ground plane, as depicted in Figure 10a. By introducing a horizontal stub to the right side of the ground plane slot and a vertical stub to the top left of the feedline, a slight improvement was achieved for the lower and upper-frequency bands. Meanwhile, the AR values were greatly affected, dropping from 30 dB to 2 dB at the middle frequencies. This enhancement was attributed to the stub supporting the horizontal electric component. A new technique was proposed for broadening the IBW and ARBW; this technique involves shifting the microstrip feed line to the right edge of the rectangular slot. Figure 10b,c show the rectangular slot antenna with measured IBW and ARBW of 90.2% (3.5–9.25 GHz) and 40% (4.6–6.9 GHz), respectively. Although the new technique can expand CP bands, it realises a narrow ARBW with an extremely low overlapping ratio of 40%.

In Xu, Li and Liu [11], a broadband CP slot antenna was designed with a sword-shaped radiator patch and a C-shaped slot. As shown in Figure 11, two arc-shaped stubs are etched into the left side of the rectangular slot to change its shape into a C-shaped slot. This method supports horizontal and vertical electric components whilst keeping a 90° PD between them to improve the CP band. The two arc-shaped stubs are cut on the left edges of the rectangular radiator to convert it into a sword-shaped radiator patch to enhance the impedance matching and then broaden the IBW. The design achieves broad IBW and ARBW of 132% (0.85–4.15 GHz) and 95.7% (1.2–3.4 GHz), respectively. However, the band extensions were performed by extending the antenna dimensions, thereby leading to a bulky structure (Figure 11). This drawback limits the use of this antenna design in modern wireless communication systems. Other current works on BCPPSA can be found in [88].

Figure 12a shows the square CP slot antenna based on a G-shaped feed line proposed by Oteng Gyasi et al. [89]. A new impedance transformation method was used in this work by cutting one side of the feed line and introducing a small horizontal slot in the near middle portion of the feed line to enhance the impedance matching between the G-shaped radiator and the 50 Ω feeding port of the antenna. To extend the IBW, the authors embedded an inverted L-shaped slot in the lower slot ground plane. Meanwhile, the ARBW was improved by introducing a U-shaped parasitic strip on the same side of the slot ground plane. This enhancement was due to the perturbation of the current distribution on the slot ground plane after placing the parasitic strip. As revealed in Figure 2 and Figure 10a, this antenna has a compact size but provides relatively narrow IBW and ARBW (Figure 12b,c), respectively).

Ullah and Koziel [90] aimed to cope with the recent challenge in broadband CP antennas, that is, the excitation of two electric field components whilst ensuring the simplicity and compact size of the broadband CP antenna. The authors employed three new techniques. Firstly, a new feeding technique was used by coupling a bracket-shaped parasitic strip with a simple feed line (Figure 13). Secondly, two layers of the ground plane were used as the top layer to perform CPW feeding by printing the CPW ground plane on the top layer. The ground slot for the bottom layer was realised by etching a rectangular slot on the bottom conductor layer. Thirdly, an asymmetric method was employed by increasing the length of the right CPW ground plane (Figure 13).

With these techniques, in addition to a previous method involving the attachment of a horizontal strip to the left side of a slot ground plane, compact antenna size and circular polarisation were realized. Symmetrical bidirectional radiation patterns were also achieved, in addition to a stable gain ranging from 2 dBi to 3.7 dBi for the ARBW. However, as revealed in Figure 13a,b, the proposed antenna has relatively narrow bandwidths and is thus still unable to overcome the main drawback of broadband CP monopole antennas with regard to obtaining a broad CP bandwidth and compact size.

Hence, Gyasi et al. [91] proposed a new antenna structure to overcome the aforementioned limitation by designing a broadband CP antenna with an IBW and ARBW broader than 80% whilst maintaining a compact antenna size. The proposed design consists of a simple F-shaped radiator and linked rectangular and inverted L-shaped slots in the ground plane conductor (Figure 14). Additionally, a bidirectional pattern is observed with RHCP in the +z direction and LHCP in the –z-direction over the whole bandwidth. In order to achieve a unidirectional pattern, a metallic reflector could be employed at a λ/4 distance from the antenna, as shown in Figure 14c.

The use of an asymmetric pair of rectangular and L-shaped slots on the back side of the ground plane is important in impedance matching and in broadening the IBW. This enhancement is brought about by the vertical part of the L-shaped slot under the feed line that reduces the coupling between the ground plane and the feed line. The horizontal part of the L-shaped slot provides a current path. The achieved IBW was 92.6% (2.9–7.9 GHz). The CP band was improved with two horizontal stubs attached to the feed line, resulting in an F-shaped feed line. These two stubs supported the horizontal electric field component by breaking the uniformity of the current distributed on the feed line. The research successfully broadened the IBW and ARBW with values of 92.6% (2.9–7.9 GHz) and 83.5% (3–7.3 GHz), respectively. The overlapping ratio was 83.5%. However, as revealed in Table 2, the trade-off factors (IBW/size) and (ARBW/size) with values of 3.43 and 3.09, respectively, are still relatively low because the bands of the proposed antenna are centred at a high frequency of 5.15 GHz. The technical specification and working principle of this antenna have been studied widely in the literature [92,93,94,95]. Recently, two research works proposed slot antennas with different structural geometries based on radiator patch and ground plane shapes.

In the first study, Tran, Nguyen-Trong and Abbosh [96] proposed a new technique to broaden the IBW using a C-shaped radiator whilst inserting a semi-circular (C-shaped) slot in the ground plane patch (Figure 15). To broaden the ARBW, the authors applied an asymmetric CP slot antenna, i.e., they shifted the inverted L-shaped feed line and the semi-circular slot to the edge of the ground plane to generate circular polarisation and broaden the IBW. Although broad bandwidths were achieved, these improvements were obtained by extending the antenna dimensions and by degrading the antenna gain; the achieved CP gain was 1–3.2 decibel-isotropic for a circularly polarised antenna (dBic). All the measurement results of the proposed antenna are listed in Table 2 and Table 3. Other recent works by [88,97,98] used the same technique of etching a semi-circular slot in the ground plane with slight differences.

Wang et al. [99] proposed a new shape for the radiator by connecting the tapered edge of a modified inverted L-shaped patch to the feed line. An impedance transformation was applied to match the 50 Ω feeding port with the radiator patch. To enhance the IBW, the authors added two triangular stubs and two vertical stubs to the square ground plane slot and etched two rectangular slots in the vertical borders of the square ground plane slot. Spiral and L-shaped strips were also embedded into the ground slot to broaden the ARBW. However, Figure 16 shows that the authors used a complicated structure to realise a broad IBW and ARBW with an extremely low overlapping ratio and trade-off factors of (IBW/size ratio) and (ARBW/size ratio). Further details are listed in Table 2 and Table 3. Figure 16b presents the measured and simulated normalized radiation patterns in the yz-plane and xz-plane at 5.9 GHz.

Ullah and Koziel [24] presented a simple coplanar strip technique with CPW feeding for the design of slot CP antennas for broadband wireless communication systems. The technique involves a simple straight radiator and two inverted L-shaped parasitic strips placed coplanar to the straight monopole (Figure 17). Strong vertical and horizontal electric field components were realised due to the excitation of the current distribution on the straight monopole and ground slot. The IBW and ARBW were thus improved.

Two techniques were employed to further improve the ARBW. The first one involves the addition of a horizontal strip to the right vertical border of the ground slot. This horizontal strip offers a current path in the horizontal direction, thereby inducing another horizontal electric field component (EH-field). The second technique involves increasing the length of the right CPW ground plane to achieve an asymmetric CPW ground plane and in turn mitigate the current cancellation produced from the opposite directions of the currents on the CPW ground planes.

The simulation and measurement results showed good agreement. As shown in Figure 17, the measured IBW and ARBW are 72% (2.2–4.85 GHz) and 66% (2.4–4.87 GHz), respectively. The measurement results indicated that the proposed antenna has broad IBW and ARBW with a high overlapping ratio. However, the response bands need to be extended towards the lower and upper frequencies to cover most wireless communication bands, such as GSM and the lower and upper WLAN and WiMAX bands. The wireless device works at 5.15–5.35 and 5.725–5.825 GHz. Several researchers have focused on the development of a CPW-fed BCPPSA, including its working principle, technical specifications and analysis [100,101].

The latest development in broadband CP slot antennas was proposed by Huang and Zhang [18], who used different IBW and ARBW improvement techniques. The proposed design integrates most of the techniques used in previous research as shown in Figure 18. For example, a coupler parasitic strip was used to extend the CP band towards the lower and upper frequencies as employed by Ullah and Koziel [90], and linked slots in the ground plane patch were used to expand the IBW as employed by [91]. In addition, the authors used the same asymmetric ground plane technique for ARBW improvement as that used by Ullah and Koziel [102], and a vertical slit in the ground plane under the feed line was provided to reduce the coupling between the ground plane and the feed line and thereby enhance the impedance matching as applied in [103]. An inverted L-shaped strip was also connected to the ground slot edge to further broaden the ARBW.

Figure 1 shows that the proposed antenna offers broad IBW and ARBW of 100% (2.15–6.44 GHz) and 108% (2.1–7 GHz), respectively, with high overlapping ratios that reach 86%. Moreover, the proposed antenna covers most wireless communication frequencies, excluding those operating at less than 2 GHz. Although this CP antenna achieves broad IBW and ARBW, these band extensions were performed by extending the antenna dimensions (45 × 43.6 mm^2^). Moreover, the trade-off factors (IBW/size ratio) and (ARBW/size ratio) are limited. These drawbacks limit the use of this antenna in modern compact communication systems.

Several types of slot antennas have been developed to analyse the suitable methods for realising broadband CP antennas. Various techniques and improvements have also been presented for reducing antenna size whilst retaining wide IBW and ARBW. Table 3 chronologically summarises the specifications of BCPPSAs for the period of 2015–2020. The materials used for each antenna design are also detailed. Table 2 provides the measurement results for the antenna designs presented in Table 3.

The comparison of the proposed designs in Table 2 is performed in terms of IBW, ARBW, antenna size, ratio of IBW and ARBW to size, overlapping percentage and antenna gain. The centre frequency Fc of the IBW is used to calculate the free-space wavelength (λo) and antenna gain in units of dBi and dBic (as presented in the sources); here, the antenna gain refers to the measured gain achieved over the entire ARBW. The size is presented in millimetres (mm) and wavelength (λo^2^). For a recent trade-off analysis, three terms are introduced in the last three columns to achieve a fair comparison. The first two terms cover the trade-off simplicity between the achieved IBW and ARBW and the antenna size. The last column shows the overlapping percentage between the achieved total IBW and ARBW.

**Table 2 micromachines-13-01048-t002:** Comparison of Existing BCPPSAs.

#	Ref	*fc* (GHz)	IB-W (%)	AR-BW (%)	Gain (dB), (dBi), (dBic)	Size (mm^2^), λo^2^	IBW/Size	ARBW/Size	OL%
1	[74]	2.84	64	48.3	2.5–3.9 dB	60 × 50, 0.27	2.37	1.79	77
2	[85]	5.82	84	41	3–3.5 dBi	28 × 28, 0.3	2.8	1.37	48
3	[86]	6.375	90.2	40	0.8–4.5 dBi	25 × 25, 0.28	3.2	1.43	40
4	[87]	2.5	132	95.7	3–4.7 dBi	100 × 100, 0.69	1.9	1.38	67
5	[89]	5.1	70.7	37.1	1.7–3.8 dBi	30 × 30, 0.26	2.72	1.43	56
6	[90]	5.2	62	49	2–3.7 dB	27 × 28.8, 0.23	2.63	2.13	72
7	[91]	5.15	92.6	83.5	2–5 dBi	30 × 30, 0.27	3.43	3.09	84
8	[96]	4.5	104	91	1–3.2 dBi	42 × 42, 0.4	2.62	2.27	76
9	[97]	8.6	125	61	4–4.3 dBi	25 × 25, 0.51	2.45	1.2	26
10	[99]	9.6	92.7	54.2	2–4.5 dBi	40 × 40, 1.6	0.56	0.34	48
11	[18]	4.3	100	108	2.3–5.5 dB	45 × 43.6, 0.4	2.5	2.7	86
12	[102]	3.52	72	65	2–6.3 dBic	37 × 39, 0.2	3.6	3.25	93
13	[103]	4.5	84.1	68.7	1.8–4 dBi	30 × 30, 0.2	4.2	3.4	85
14	[104]	1.71	63.2	36.8	0.3–3.2 dBi	60 × 60, 0.34	1.86	1.1	65
15	[105]	4.91	106	89	1.9–5.2 dBi	40 × 45, 0.48	2.2	1.85	70
16	[106]	5.7	117	59	1– 3.2 dB	38 × 38, 0.52	2.25	1.13	41

Abbreviations: AR, axial ratio; ARBW, axial ratio bandwidth; IBW, impedance bandwidth, OL, overlapping.

**Table 3 micromachines-13-01048-t003:** Summary of BCPPSAs from the Literature.

Ref	Material	Radiator Shape	Technique to Broaden IBW	Technique to Generate the CP Property
[74]	FR4	Asymmetric L-shaped	Decreasing and increasing the vertical and horizontal arms of the L-shaped radiator, respectively	Placing a rectangular parasitic patch above the L-shaped radiator and surrounding the exterior borders of the patch with a rectangular strip
[85]	Roger RT/5880	U-shaped	Connecting the Y-shaped strip to the square slot ground	Employing half of the Y-shaped radiator and small strips as a parasitic strip
[86]	FR4	Sword-shaped patch	Chamfering the two left corners of the radiator patch	Placing two arc-shaped stubs on the left corners of the ground plane slot
[87]	FR4	Straight with vertical stub	Shifting the feed line to the right of the rectangular slot	Connecting the horizontal stub to the rectangular slot and shifting the feed line
[89]	FR4	G-shaped patch	Inserting an L-shaped slot in the lower part of the ground plane slot	Introducing a U-shaped parasitic strip into the square slot
[90]	Arlon AD250C	Straight	Introducing a simple horizontal strip into the slot ground plane	- Coupling the bracket-shaped parasitic strip to the straight feed line - Employing an asymmetrical ground plane
[91]	FR4	F-shaped	Inserting an inverted L-shaped slot in the lower border of the ground plane slot	Attaching two horizontal stubs to the feed line
[96]	Taconic RF-35	C-shaped	Using an optimised C-shaped radiator	Etching a C-shaped slot in the ground plane
[97]	FR4	Inverted L-shaped	- Shifting the feed line to the edge of the substrate - Using the inverted L-shaped radiator	- Shifting the semi-circular slot and feed line to the edge of substrate - Inserting a small horizontal slit in the ground plane
[24]	Arlon AD250C	Straight feed line	- Shifting the feed point and using optimised square slot dimensions - Coupling the straight monopole with two inverted L-shaped strips	- Attaching a horizontal strip to the edge of the ground slot - Employing an asymmetric CPW ground plane - Coupling the straight monopole with two inverted L-shaped strips
[18]	FR4	Straight feed line	Etching three linked slots in the ground plane patch	- Coupling the feed line with a bracket-shaped parasitic strip - Applying an asymmetric ground plane
[103]	FR4	Modified straight feed line	Modifying the feed line by attaching an L-shaped strip and a rectangular slot in the lower part of the feed line	Adding a cross-shaped parasitic strip in the middle of the ground slot
[104]	FR4-epoxy	Asymmetric Y-shaped	Attaching a vertical rectangular stub to the right side of the feed line	Connecting two inverted L-shaped strips to the two corners of a ground plane slot in diagonal directions and modifying the borders of the ground plane slot
[105]	FR4	Modified L-shaped patch	Embedding a meandered arc-shaped slot into the radiator patch	- Etching small horizontal and L-shaped slots in the lower side of the ground plane - Cutting the four corners of the ground plane
[106]	Taconic RF-35	Semi-circular patch	Using a conventional semi-circular radiator	Etching a full circular slot in the ground plane

Figure 19 shows the analysis process of the BCPPSA performances in Table 2. Sixteen research works for the BCPPSAs are presented in Table 2. As revealed, only eight antennas have trade-off factors higher than 1.5 as in [1,6,7,8,11,12,13,15]. According to the performance results in Table 2, the percentage of BCPPSAs with high trade-off values is 50%. In contrast, only 6 BCPPSA designs, 37.5%, offer wide overlapping IBW and ARBW of more than 75% (overlapping percentage > 75%) while maintaining a high trade-off value as in [1,7,8,11,12,13]. Only 2 antenna designs, 12.5%, offer fractional bandwidths (IBW and ARBW) of more than 90% while maintaining a high trade-off value and a wide overlapping band as in [8,11]. However, these have low CP antenna gain with a big variation over most of the achieved CP band as in [8], or the antenna size is quite large as in [11]. However, the ratio of the slot antenna designs offering a broad IBW and ARBW while retaining high trade-off factors and overlapping percentage is 18.75%.

Moreover, Figure 19 indicates that all existing BCPPSAs are unable to offer the previous characteristics with compact size (λo^2^ < 0.3) and a small variation over most of the achieved CP bands. The weak gain in these studies may be attributed to the operation of the antennas at low frequencies, as in [14]. Moreover, the techniques used to broaden the bandwidths of the antennas whilst keeping the antennas compact may contribute to the deterioration of CP antenna gain, as shown in [9].

In this section, related studies are reviewed along with the unique methods or techniques they introduced to develop broadband CP antennas. The most sought-after characteristics of broadband CP antennas are broad IBW and ARBW, the high overlapping percentage between IBW and ARBW and the compact size and good antenna gain over the majority of the achieved CP band. Extensive efforts have been exerted to realise a BCPPSA design possessing all the aforementioned characteristics. However, slot antenna designs (Table 2) with broad IBW and ARBW, high overlapping percentage of >75% and compact size are rare. Therefore, the demand for new broadband CP antennas with compact sizes, broad operating bands, wide overlapping bands and the high trade-off between antenna size and bandwidths has increased.

### 4.2. Broadband CP Printed Monopole Antennas (BCPPMAs)

One of the most popular antennas used in modern wireless communication systems is the monopole antenna. The conventional monopole antenna consists of a vertical wire, a helical whip or a tube which is constructed above the ground plane [69]. The length of a monopole antenna is typically around λo/4 as a function of the wavelength of its resonant frequency [107,108,109]. The radiation pattern produced by a monopole antenna is similar to that produced by a dipole antenna but only in the half-space above the ground plane. Hence, the directivity of a quarter-wave monopole above a ground plane is twice that of a half-wave dipole radiating in free space [110].

Antenna designers have focused on monopole antennas for use in wireless communication systems due to their advantages such as omnidirectional coverage, simple structure, low cost and capability of offering wideband, multi-band and broadband operations [111,112,113]. However, monopole antennas have large sizes, heavy profiles and pure vertical polarisation due to their weakness in the horizontal component of far-field radiation [114]. As a solution to the large size limitation and the need to maintain the attractive features of monopole antennas, printed monopole antennas have become particularly attractive as they combine the desirable characteristics of monopole antennas and printed antennas, including their simple structure, omnidirectional radiation pattern in the horizontal plane, straightforward design and manufacture, ease of broadening operation bands and capability of generating circular polarization [115,116].

Printed monopole antennas are those whose feed line, radiating patch and ground plane are oriented on the same plane. These conducting components are etched on a dielectric material plane (substrate) with known properties and dimensions. These antennas fall into two different configurations in terms of the location of the ground plane, feeding line and radiator patch; that is, they can be microstrip-fed printed monopole antennas (Figure 20) [117] or CPW-fed printed monopole antennas [16] (Figure 21).

To address the relentless demand for minimising the number of indoor antenna elements within an indoor environment, designers have focused on BCPPMAs with circular polarisation that can cover most wireless communication bands. Modern BCPPMAs combine the desirable features of monopole and printed antennas as well as their circular polarisation. In developing these antennas further, one must know that the pure polarisations of printed monopole antennas are linear (vertical) because of the weakness in their horizontal radiation. Generating CP radiation requires an equal amplitude and a 90° PD for two electric field components in the vertical (EVer) and horizontal (EHor) directions [118,119]. Therefore, a number of applicable techniques are needed to design BCPPMAs.

Recently, numerous BCPPMAs based on different techniques have been proposed in response to the need to excite two electric field components with equal amplitude and a 90° PD so as to generate the CP property whilst maintaining the required compact size, broad IBW and acceptable gain value for the achieved CP band. However, the main drawback of the BCPPMAs is the difficulty of obtaining a broad IBW and broad CP bandwidth with their compact size [120,121,122].

As a solution to the difficulties in the trade-off between antenna bands and antenna size, several BCPPMAs have been proposed in the literature. These proposed antennas include a C-shaped monopole antenna [123], whose fundamental response is produced by a C-shaped radiator and a rectangular ground plane. Single impedance transformation is also employed to match the 50 Ω feed line and radiator by adding two triangular stubs to the rectangular ground plane. As shown in Figure 22a, two strong electric components (EVer, EHor) are generated, and the IBW is enhanced by the addition of a half-ellipse strip and an extension stub on the upper and lower ends of the C-shaped radiator, respectively. A slit is inserted in the lower part of the radiator, and a 90° PD between EVer and EHor is achieved. Thus, a CP band is produced with a broad ARBW that extends up to 104% (2.05–6.55 GHz). The proposed antenna can offer a broad IBW of 106% (2.25–7.35 GHz) and a bidirectional radiation pattern with a relatively large isolation of 10 dB between the RHCP and the LHCP, as shown in Figure 22c. However, the broad bandwidths achieved by the proposed antenna compromised the antenna dimensions, with the trade-off value being extremely low (Table 4). Other studies have explored CP monopole antennas with similar radiators and ground shapes [124,125,126].

Ding, Guo and Gao [127] presented the simple CPW-fed monopole antenna shown in Figure 23a. In general, the horizontal components of the CPW ground plane of the CPW-fed monopole antenna are in opposite directions, as shown in Figure 23a. Therefore, the horizontal surface current in the ground plane is counteracted, and the radiation of the conventional CPW-fed monopole antenna under far-field conditions is vertically polarised (VP). Hence, the authors employed an asymmetric ground plane with CPW ground planes with different high gains to mitigate the cancellation in the horizontal component and then generate the fundamental CP band at the low frequencies at which the AR values decrease.

To further expand the ARBW and IBW, a square open-loop parasitic strip is employed which is placed beside a rectangular radiator. The ARBW is broadened because the open loop is capacitively coupled with the rectangular radiator patch, which transmits energy to the open loop under the travelling wave model. Meanwhile, the IBW is expanded due to the increase in the current paths after the radiator is coupled with the open-loop parasitic strip. As observed in Figure 23b,c, the proposed design offers wide IBW and ARBW of 96.5% (1.48–4.24 GHz) and 63.3% (2.05–3.95 GHz), respectively. However, the CP band needs to be enhanced to cover most wireless communication systems. Moreover, the measured CP antenna gain suffers from a large variation within the range of 0.5–3.5 dBi. In sum, the proposed monopole antenna has relatively wide bandwidths and bulky size, and its size is an obvious compromise. Hence, the use of this antenna is limited to compact wireless devices.

Chen et al. [128] proposed a different method for mitigating the cancellation of surface currents on the CPW ground plane and support the horizontal component without the need to increase or decrease the length of any part of the CPW ground plane. The method involves a horizontal slit and an inverted L-shaped strip on the right-hand side of the CPW ground plane, as shown in Figure 24. The horizontal slit inserted into the ground plane disturbs the surface current and changes its direction. The coupled effect between the asymmetric ground plane and the inverted-L strip plays an important role in supporting the horizontal and vertical components. This design achieves wide ARBW and IBW of 58.8% (4.8–8.8 GHz) and 47.8% (5.37–8.75 GHz), respectively. However, despite its compact size, the bandwidths need to be broadened to cover most wireless communication frequencies. Other current works on BCPPMAs with an inverted L-shaped technique were reported by [129].

The CP monopole antennas presented by [127,128] employ an asymmetric ground plane by changing the length of the CPW ground plane or by introducing strips and slits in the CPW ground plane. These methods present an effective solution for generating CP bands for conventional CPW-fed monopole antennas. The technical specifications and working principles of these antennas have been studied widely in the literature [130,131,132,133].

Microstrip-via presents another solution to the main drawback of CP monopole antennas as it can increase the horizontal component to achieve equilibrium with a vertical current distribution [134]. In this design, a spiral strip is placed on the edge of the left-ground plane. The end of the spiral strip is connected to a rectangular radiator through a via, as shown in Figure 25. A circumferential horizontal current is then produced in the spiral strip, along with strong horizontal and vertical currents on the left CPW ground plane and radiator. Circular polarisation is thereby generated. However, the CP band remains narrow.

After employing the conventional technique for generating circular polarisation, two diagonal corners of the rectangular radiator are cut, and the ground plane is designed to have different heights. In this way, broad ARBW is achieved. A vertical slot is inserted in the ground plane located under the feed line, thereby affecting the IBW greatly due to the reduced coupling between the feed line and the ground plane. Although this antenna achieves broad IBW and ARBW, it is relatively large, its trade-off value is low (Table 5) and the measured antenna gain varies greatly within the CP band (ARBW); the peak gain is 4 dBi at 4.1 GHz, whilst the lowest gain is 0.5 dBi at 5 GHz. Figure 25c reveals that the proposed antenna provides a bidirectional radiation pattern (LHCP and RHCP). The figure also indicates the poor isolation of less than 10 dB in the xoz plane at 4.7 GHz between the LHCP and RHCP patterns. This impairment is attributed to the via technique or the use of an asymmetric ground plane. The technical specification and working principle of this antenna have been studied widely in the literature [135].

With the same objective, Lv and Yang [136] employed two linked inverted L-shaped spiral strips with four triangular stubs added to the corners to generate a CP wave without using the microstrip via method. Despite its broad bandwidths, the proposed antenna is still relatively large. As reported by Jhajharia et al. [137], a shorting pin was used to connect the horizontal arm of a radiator patch to a rectangular stub on a modified ground plane to increase the physical dimension of the horizontal arm and thereby provide another path for the current rotation. However, the use of a shorting pin to expand the CP band results in a low isolation between the achieved bidirectional radiation patterns. Hence, the application of this technique to microstrip indoor antennas is limited because of the importance of bidirectional radiation patterns in IWC systems [21].

Samsuzzaman, Islam and Singh [138] proposed a CP monopole antenna, as shown in Figure 26a. A hook-shaped branch is connected to a partial ground plane to excite the surface current and thereby broaden the CP band. The gap between the partial ground plane and the asymmetric inverted L-shaped feed line works as a coupling capacitance, which plays an essential role in IBW improvement. As shown in Figure 26b, the simulation and measurement results of the proposed antenna show a good agreement. However, the achieved IBW and ARBW are less than 70%. Moreover, as observed in Figure 26a, despite the antenna’s simple design, the authors did not attempt to boost the bands by using simple techniques such as slots or strips in the ground plane.

A CP monopole antenna with a modified inverted C-shaped radiator patch is proposed by Midya, Bhattacharjee and Mitra [139] as shown in Figure 27a. A single impedance transformation is applied to enhance the impedance matching between the 50 Ω feed line and the radiator patch. The inverted C-shaped patch is optimised by decreasing the upper arm length of the rectangular C-shaped patch so as to generate the first response. A G-shaped parasitic strip is introduced into the radiator patch cavity to perturb the surface current on the main radiator and to offer other current paths to support the horizontal and vertical components and thereby generate the first CP band. Meanwhile, the minima of the IBW are left shifted towards the low frequencies.

Two symmetric rectangular slots in the ground plane perturb the surface current on the ground plane and balance the vertical and horizontal electric field magnitudes to make them equal to a 90° PD for CP wave generation and for further broadening the ARBW. As presented in Figure 27b,c, the proposed antenna offers broad IBW and ARBW with a good agreement and a high overlapping percentage. As observed in Figure 27c, the CP gain is high, in the range of 2.5–3.5 dB. However, despite the proposed antenna’s relatively compact size, its trade-off value is low, as illustrated in Table 5. This drawback is attributed to the achieved IBW and ARBW being centred at a high frequency. Other current works on BCPPMAs that used the same technique of cutting the upper corners of the ground plane were reported by Sam and Abdulla [140] but with different slot shapes (triangular slots).

In another research work, Li et al. present a miniaturised BCPPMA with broad bandwidths and a small size [141]. As depicted in Figure 28, a new technique is used to produce the radiator patch. This technique involves the combination of circular and square rectangular ring structures. The combination provides a fundamental wideband IBW and a narrow CP band at low frequencies before modifying the ground plane or adding a parasitic strip. To broaden the bandwidths, the authors place a triangular stub on the left corner of the ground plane and cut the lower corner of the C-shaped radiator patch. As presented in Table 5, the proposed antenna has broad IBW and ARBW, compact size and high CP gain. The highest overlapping percentage for the ARBW and IBW is 96%. However, the trade-off values of IBW/size and ARBW/size for the proposed antenna are considerably low at 1.94 and 1.97, respectively. This weakness is attributed to the IBW and ARBW being centred at a high frequency of 8.4 GHz. Therefore, the application of this antenna, particularly to GSM 1800, GSM 1900 and LTE 2400, is limited.

Another BCPPMA structure was proposed by Ellis et al. [142]. As shown in Figure 29a,b, the design of the BCPPMA is relatively simple because it does not use slots, stubs or parasitic strips. The proposed antenna only consists of an asymmetric-fed (lateral feeding) rectangular radiator patch and an inverted L-shaped ground plane. By optimising and right shifting the rectangular patch to form asymmetric feeding, dual S11 responses are achieved at the lower and upper frequencies. As a result of the shifting of the feed line and the radiator patch to the left edge of the substrate, S11 is tremendously improved, and the first CP band is produced. The rectangular ground plane is modified by etching a horizontal rectangular slot in the main ground plane to produce a laterally inverted L-shaped ground plane. This design boosts the IBW and ARBW.

Figure 30a,b show that the proposed antenna can offer an IBW of 88.9% and an ARBW of 66.7%. Although it has a simple and compact structure, the range of the gain values within the CP band is low (3–4 dBic). As shown in Figure 30c, the ARBW requires further expansion so that it overlaps with the IBW because the overlapping ratio between them is less than 78%. Additionally, both bands can be broadened by coupling the rectangular radiator patch with an open-loop or spiral-loop parasitic strip, similar to that used in [127].

Table 4 summarises the methods and techniques used for the design of recent BCPPMAs from 2015 to 2020. The table also indicates the materials used for each antenna design. The most widely used technique for enhancing impedance matching is impedance transformation in the conventional feed line. The feed line is designed with different widths and lengths relative to the impedance used. Another method of broadening the IBW is to reduce the coupling between the ground plane and the feed line by inserting a vertical slot in the ground plane directly under the feed line or increasing the flow current on the opposite edges of the radiator patch and ground plane by keeping an optimised gap between the radiator patch and the ground plane. In broadening the ARBW, a popular technique is to couple the monopole radiator with a parasitic strip with different shapes to excite the current distribution on the radiator patch, to generate strong vertical and horizontal components or to embed slits or slots into the ground plane. The most common method of enhancing the IBW and ARBW simultaneously is to introduce stubs or strips into the ground plane. This method greatly affects antenna performance because it offers other current paths, especially in the horizontal direction, to overcome the main drawback of monopole antennas, that is, the weak radiation in the horizontal direction.

**Table 4 micromachines-13-01048-t004:** Summary of the Broadband CP Printed Monopole Antennas (BCPPMAs) from the Literature.

#	Material	Radiator Shape	Technique to Broaden IBW	Technique to Generate the CP Property
[96]	FR4	Modified C-shaped	Adding two asymmetric triangular stubs above the ground plane	Extending the lower and upper edges of the C-shaped radiator and inserting a small slit in the radiator
[127]	FR4	Rectangular	Coupling the radiator with the square ring-shaped patch using a parasitic strip	Applying asymmetric CPW ground plans
[128]	FR4	Straight monopole	Placing an inverted L-shaped strip on the ground plane	Embedding a horizontal slit into the right CPW ground
[134]	FR4	Modified rectangular	Embedding a vertical slot into the ground plane located under the feed line	Using the microstrip via method to achieve equilibrium between the vertical and horizontal components
[137]	FR4	Monopole patch with perpendicular arms	Using an optimised meandering shaped Embedding a linked X-shaped slot and rectangular slot in the ground plane	- Connecting a rectangular stub on the back substrate to the horizontal arm of the radiator patch - Employing slanting edged DGS in the ground plane
[138]	FR4 epoxy	Asymmetric inverted L-shaped feed line	Keeping an optimised gap between the feed line and the upper edge of the ground plane	Hook-shaped branch connected to the partial ground plane corner for the purpose of coupling capacitance
[139]	FR4	Inverted rectangular C-shaped	- Optimising the arms of an inverted C-shape patch - Employing impedance transformation	- Introducing a G-shaped parasitic strip to a rectangular C-shaped patch - Inserting two rectangular slots in the top left and right corners of the ground plane
[141]	FR4	Modified inverted C-shaped	Forming the monopole radiator by combining circular and rectangular rings	Modifying the ground plane by placing a triangular stub on the left side of the ground plane
[142]	FR4	Asymmetric-fed rectangular shaped	Employing an asymmetric-fed rectangular patch as the radiator patch and an edge-fed rectangular monopole	- Employing an edge-fed rectangular monopole - Changing the rectangular shape ground plane into a laterally inverted L-shaped ground plane
[143]	FR4	Chifre-shaped	Using a quarter circular patch as the radiator	Transferring the feed line and radiator towards the right
[144]	FR4	Straight monopole	- Employing a single impedance transformation - Coupling the feed line by using two spiral parasitic strips	Coupling the feed line by using two spiral parasitic strips
[145]	FR4	Cross-shaped monopole	- Attaching an asymmetric stub to the feed line - Inserting a slit in the ground plane under the feed line	Extending the right side of the ground plane to support EVer and EHor. The vertical slot in this extension yields a 90° PD between EVer and EHor
[146]	FR4	Coin-shaped	Inserting two linked slots in the radiator patch	- Attaching a vertical stub to the lower border of the radiator patch - Employing an asymmetric ground plane
[147]	FR4	Straight monopole	- Applying impedance transformation - Modifying the ground plane by using two orthogonal slits	- Placing two parasitic loops on both sides of the substrate - Introducing two air gaps in the corner of the parasitic loop to yield two open-loop shaped patches
[148]	FR4	Quasi C-shaped	Employing impedance transformation and a quasi C-shaped patch	- Placing a rectangular open loop parasitic strip above the ground plane - Connecting a vertical rectangular stub to the ground plane

**Table 5 micromachines-13-01048-t005:** Summary of the BCPPMA Measurement Results in the Literature.

#	Ref	*fc* (GHz)	IB-W (%)	AR-BW (%)	Gain (dB), (dBi), (dBic)	Size (mm^2^), λo^2^	IBW/Size	ARBW/Size	OL%
1	[123]	4.3	106	104	2–6 dBic	49 × 55, 0.55	1.9	1.9	88
2	[127]	2.9	96.5	93	0.5–3.5 dBi	50 × 55, 0.25	3.86	2.5	69
3	[128]	4.1	94	77	0.5–4 dB	50 × 48, 0.44	2.13	1.75	72
4	[134]	6.8	59	48	1.8–3.4 dB	25 × 24, 0.31	1.9	1.55	85
5	[137]	4.9	102	37.5	2–2.5 dB	32 × 39, 0.33	3	1.14	40
6	[138]	3.48	56	63.6	2–3.7 dBic	44 × 44, 0.26	2.15	2.4	79
7	[139]	5.72	62.9	53.9	2.5–3.5 dB	32 × 30, 0.35	1.8	1.54	87
8	[141]	8.4	95.2	96.8	3–6.3 dBi	25 × 25, 0.49	1.94	1.97	96
9	[142]	4.65	88.9	66.7	3–4 dBic	19.5 × 36, 0.19	4.71	3.51	77.5
10	[149]	2.9	50	49	2–2.5 dBic	46.6 × 70, 0.32	1.56	1.53	89
11	[143]	2.4	72	41.6	2.5–3.6 dBi	58.4 × 63, 0.48	1.48	0.87	58
12	[144]	3.4	88	64.7	0.5–2.5 dBic	55 × 50, 0.35	2.51	1.8	80
13	[145]	6	55.5	42.6	0.4–2.7 dBi	16 × 22, 0.14	3.96	3	77
14	[146]	5.8	89.2	71	3.5–4 dBi	50 × 50, 0.93	0.96	0.76	78
15	[147]	4.65	87.2	83.9	1–2.3 dBi	55 × 55, 0.72	1.21	1.16	92

Table 5 presents the measurement results for the recent BCPPMAs introduced in Table 4. Figure 31 shows the analysis process of the BCPPMAs’ performances in Table 5. Sixteen research works for the BCPPMAs are presented in Table 5. As revealed from Figure 31, 12 antennas have high trade-off factors of more than 1.5 as in [1,3,4,5,6,8,10,11,12,14,15]. According to the performance results in Table 5, the percentage of BCPPMAs with high trade-off values is 75%. At the same time, the number of BCPPMAs that offer wide overlapping IBW and ARBW of >75% while maintaining a high trade-off value is 9 designs, 56.25%, that match [3,5,8,9,10,11,12,14,15]. Only three antenna designs offer fractional bandwidths (IBW and ARBW) of more than 90%, maintaining a high trade-off value and a wide overlapping band as in [9,11], a percentage of 12.5%. However, they have significant variation in CP antenna gain over most of the achieved CP band as in [9], or the antenna size is quite large as in [11]. The ratio of the monopole antenna designs offering a broad IBW and ARBW while retaining high trade-off factors and overlapping percentage is 18.75%.

Moreover, Figure 31 indicates that all existing BCPPMAs are unable to offer the previous characteristics of compact size (λo^2^ < 0.3) and small variation over most of the achieved CP bands. The big variation in CP antenna gain in these studies may be attributed to the operation of the antennas at low frequencies, as in [1]. Moreover, the techniques used to broaden the bandwidths of the antennas whilst keeping the antennas compact may contribute to the deterioration of CP antenna gain, as shown in [15].

Minimising the number of indoor antenna units to cover multiple frequency bands is becoming increasingly desirable in IWC systems because doing so helps reduce the cost and complexity of systems. Therefore, designing indoor antennas has three requirements: broad IBW for modern communication systems with high data rates; compact size similar to that of printed antennas with a low profile, small size, easy fabrication and low cost; and circular polarisation instead of linear polarisation because of its advantages. Another requirement is small variation in the CP antenna gain within the achieved CP band.

The main disadvantage of conventional BCPPAs is the difficult trade-off between antenna bands and size. Unlike those in ultra-wideband antenna technology, the achieved bands (IBW and ARBW) of all broadband CP antennas are not centred at the same frequency. Hence, comparing proposed antenna designs is difficult. Therefore, this work conducts an analysis of recent trade-offs. An investigation is introduced in the results comparison tables for trade-off simplicity between the achieved IBW and ARBW with antenna size to perform a fair comparison and examine if the achieved bands with the proposed antennas are of convenient size (a good optimisation).

To determine the most appropriate topology (monopole antenna or slot antenna) for an indoor antenna, four criteria are considered. The first criterion is the superiority of the evaluated trade-off factors of IBW/size and ARBW/size. The second criterion is the superiority of the overlapping percentage between the achieved bands. The third is the ability of offering a broad fractional bandwidth (IBW and ARBW) while maintaining a high trade-off value and a wide overlapping band. The fourth criterion is the simplicity of the design structure and the material cost of prototype fabrication.

According to the performance results in Table 2 and Table 5, the percentage of BCPPMAs with high trade-off values (IBW/size and ARBW/size > 1.5) is 75%, which is greater than that of BCPPSAs at 50%. Additionally, the percentage of BCPPMAs that offer wide overlapping IBW and ARBW (>75) is 68.7%, which is greater than that of BCPPSAs at 37.5%. Moreover, the percentage of BCPPMAs with broad fractional bandwidths (IBW and ARBW) of more than 90% (IBW and ARBW > 90%) while maintaining a high trade-off value and a wide overlapping band is the same for both BCPPMAs and BCPPSAs in the research.

The percentage of BCPPSAs and BCPPMAs with low gains within the ARBW is 31.25%. Moreover, BCPPMAs are known for their omnidirectional radiation property [150,151,152]. They are also capable of radiating CP waves with two stable CP radiation senses (bidirectional radiation) from opposite sides and with good isolation, i.e., RHCP wave at the +z direction and LHCP wave on the opposite antenna side (−z), and vice versa [151,152,153,154,155].

Existing BCPPSAs achieve broad IBW and ARBW with high trade-off values, but their design tends to be complex. Table 3 and Table 4 reveal that relative to BCPPSA designs, BCPPMA designs can maximise antenna performance by using low-cost materials (FR4) in fabrication. The percentage of BCPPSA designs fabricated on FR4 substrates is around 56.25%, whereas all BCPPMAs are fabricated on FR4 substrates.

Based on the previous analysis, the current research proposes BCPPMAs with biplanar and uniplanar (CPW-fed) geometries. The required sought-after characteristics highlighted in Figure 31 are a broad IBW and ARBW, high trade-off values (IBW/size and ARBW/size), high overlapping ratio, compact size and CP peak gain of more than 2.5 dBi with slight variations over most of the achieved CP bands.

## 5. Conclusions

This paper presents a review of indoor antennas, microstrip patch antennas, broadband printed antennas and the existing relevant research on the development of BCPPSAs and BCPPMAs. Existing studies about BCPPA design are compared in terms of radiator shapes and the techniques used for expanding IBW and ARBW.

The techniques used to generate circular polarisation in nonplanar antennas and printed antennas and the techniques for designing CP antennas with broad IBW and ARBW must be understood before any BCPPA is designed. The knowledge of these techniques and the choice of materials for fabrication serve as the starting points for designing the BCPPA approaches. In this review paper, the existing studies and the limitations in the design of BCPPAs were well explained. The performances of existing BCPPSAs and BCPPMAs were also compared in terms of centre frequency (F_c_), IBW, ARBW, antenna gain and antenna size. IBW/size, ARBW and overlapping percentage are introduced for further investigation.

## Figures and Tables

**Figure 1 micromachines-13-01048-f001:**
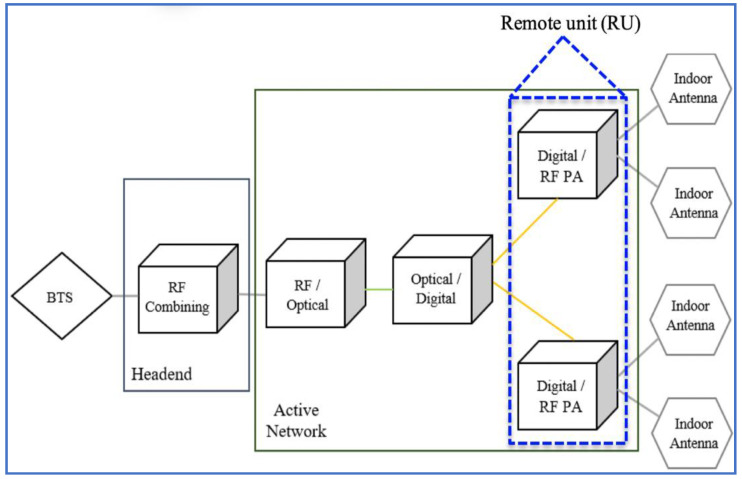
A block diagram of an indoor distributed antenna.

**Figure 2 micromachines-13-01048-f002:**
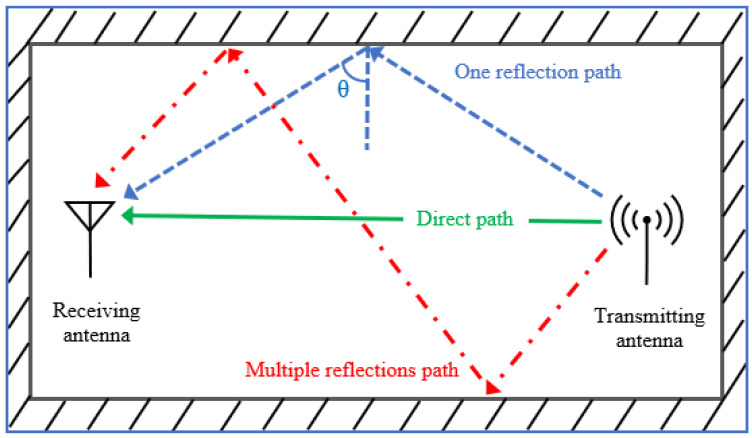
An indoor multipath propagation scenario.

**Figure 3 micromachines-13-01048-f003:**
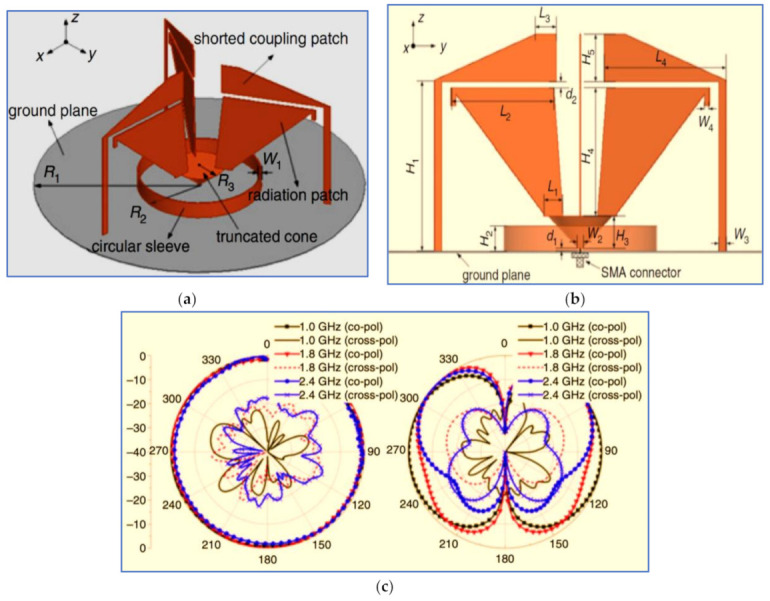
Configuration of an indoor antenna: (**a**) 3D view, (**b**) side view, and (**c**) radiation patterns at 1.5, 1.8 and 2.5 GHz [45].

**Figure 4 micromachines-13-01048-f004:**
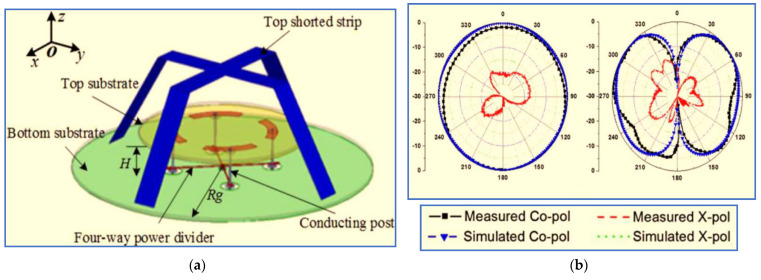
Overview of the proposed indoor antenna: (**a**) geometry and (**b**) radiation pattern at 2.0 GHz [47].

**Figure 5 micromachines-13-01048-f005:**
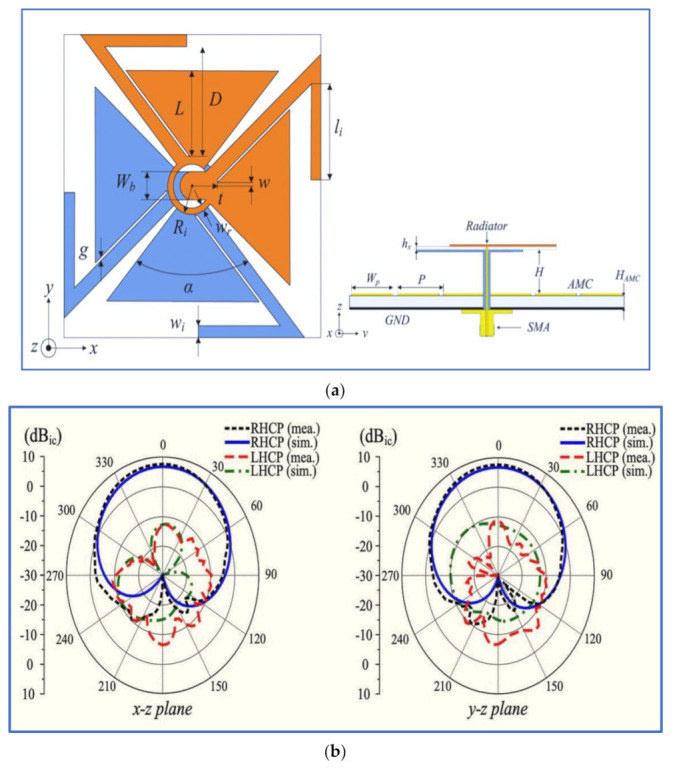
A proposed CP antenna: (**a**) geometry and (**b**) radiation pattern at 2.4 GHz [51].

**Figure 6 micromachines-13-01048-f006:**
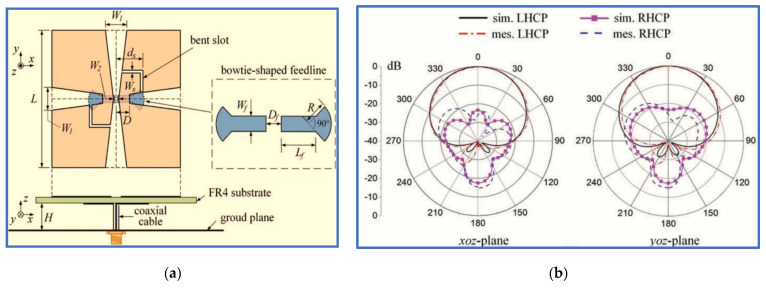
A proposed antenna: (**a**) antenna configuration and (**b**) radiation pattern at 2.0 GHz [15].

**Figure 7 micromachines-13-01048-f007:**
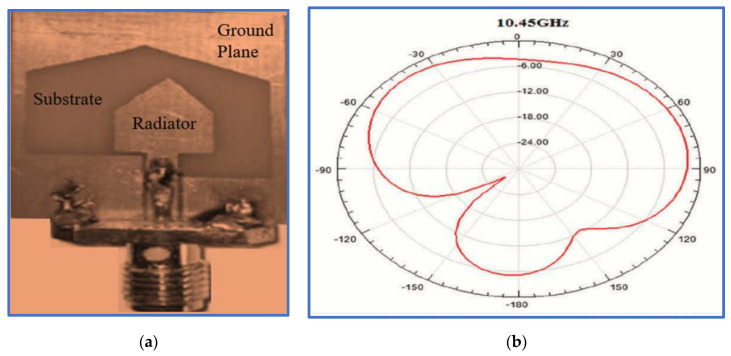
A uniplanar (CPW-fed) slot antenna: (**a**) geometry and (**b**) radiation pattern at 10.45 GHz [81].

**Figure 8 micromachines-13-01048-f008:**
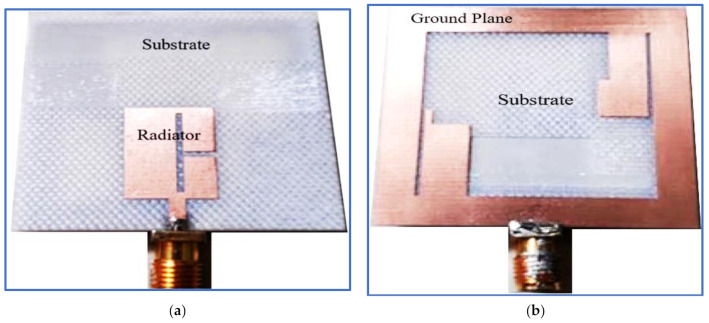
A biplanar slot antenna: (**a**) front view; (**b**) back view [82].

**Figure 9 micromachines-13-01048-f009:**
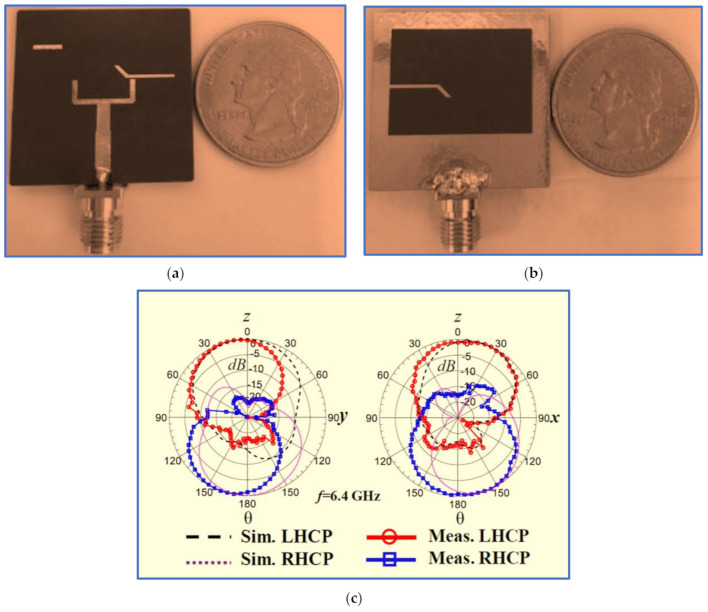
A square slot antenna: (**a**) front view; (**b**) back view; (**c**) simulated and measured radiation patterns [85].

**Figure 10 micromachines-13-01048-f010:**
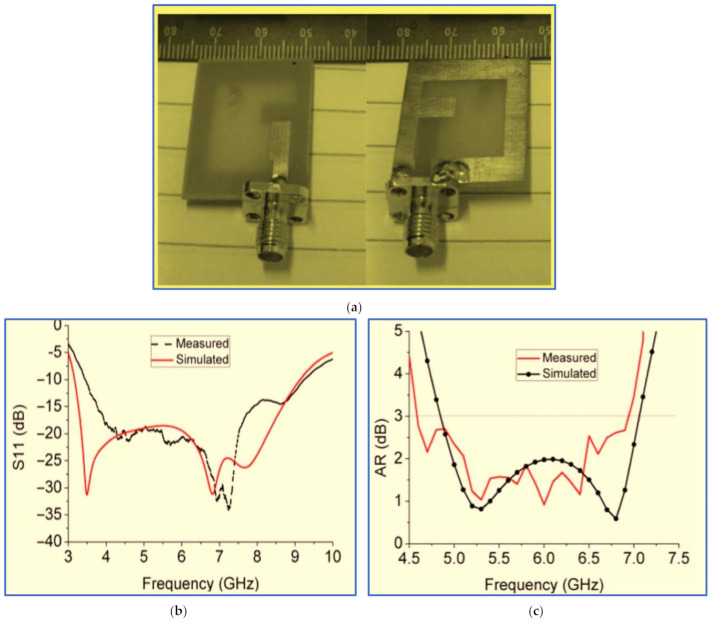
(**a**) Front and back views of a wide rectangular slot antenna. (**b**) Simulated and measured S11 results and (**c**) simulated and measured AR [87].

**Figure 11 micromachines-13-01048-f011:**
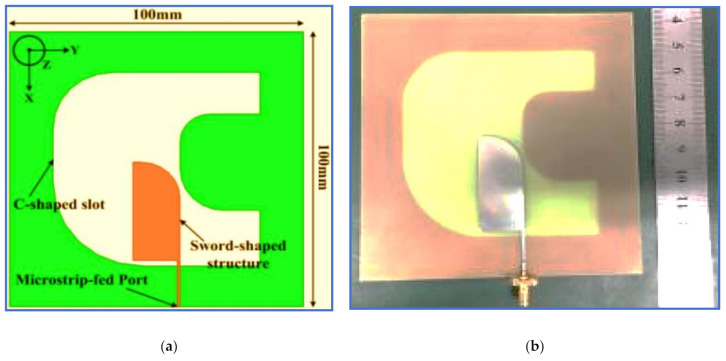
Geometry of a C-shaped slot antenna: (**a**) the dimensions of the proposed antenna and (**b**) the fabricated design [11].

**Figure 12 micromachines-13-01048-f012:**
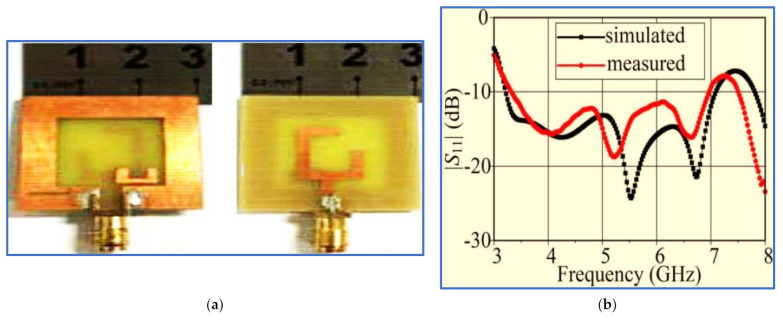
(**a**) Front and back views of a square CP slot antenna. (**b**) Simulated and measured S11 results. (**c**) Simulated and measured AR results [89].

**Figure 13 micromachines-13-01048-f013:**
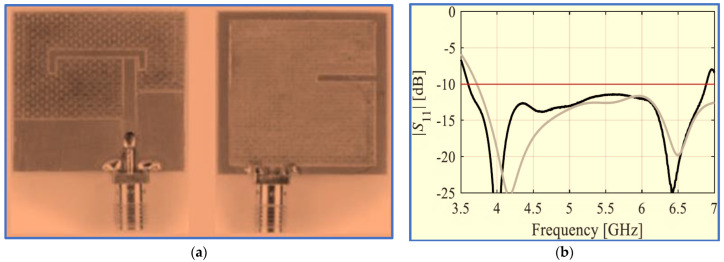
(**a**) Front and back views of a wide rectangular CP slot antenna. (**b**) Simulated (gray) and measured (black) S11 results. (**c**) Simulated (gray) and measured (black) AR results from [90].

**Figure 14 micromachines-13-01048-f014:**
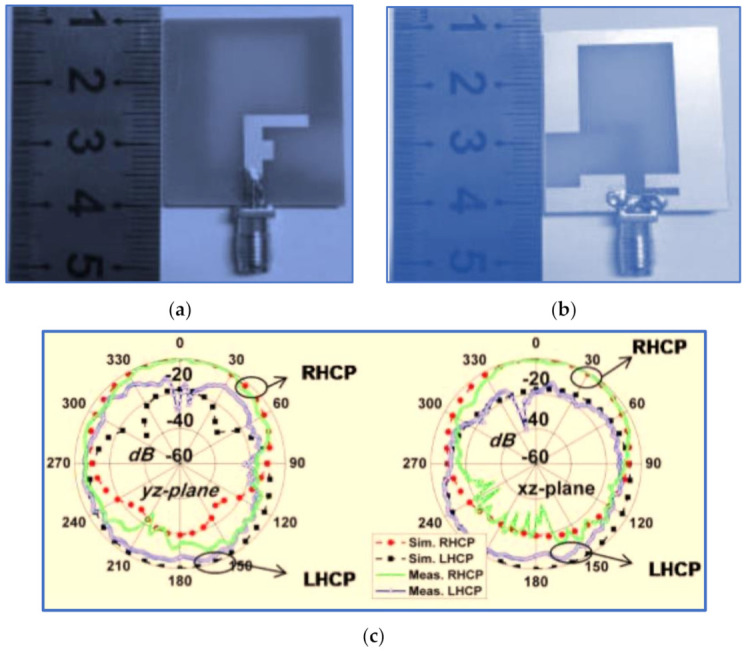
Fabricated prototype: (**a**) front view, (**b**) back view, and (**c**) simulated and measured radiation patterns at 3 GHz [91].

**Figure 15 micromachines-13-01048-f015:**
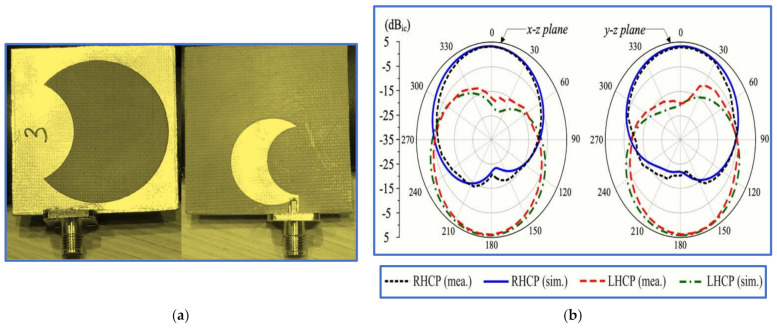
A proposed C-shaped slot antenna: (**a**) the fabricated prototype; (**b**) the simulated and measured realized gain radiation patterns at 3 GHz [96].

**Figure 16 micromachines-13-01048-f016:**
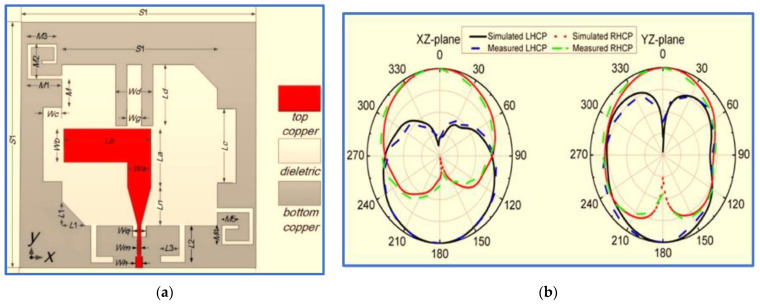
Configurations of the finalized antenna. (**a**) Configuration of inverted L-shaped slot antenna. (**b**) Simulated and measured antenna radiation patterns at 5.9 GHz [99].

**Figure 17 micromachines-13-01048-f017:**
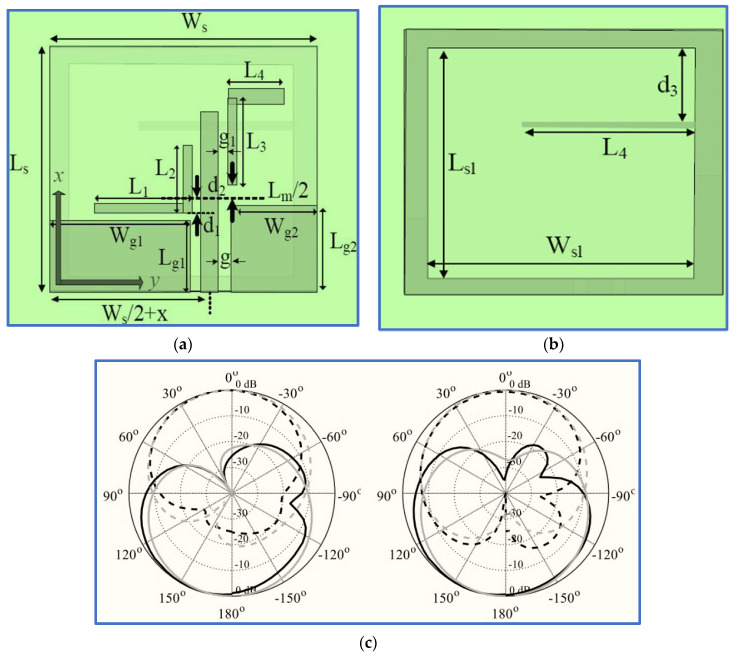
Geometry of a CP slot antenna with coplanar strip: (**a**) front side; (**b**) back side; (**c**) simulated and measured CP radiation pattern in the xz-plane RHCP and LHCP at 2.5 GHz [24].

**Figure 18 micromachines-13-01048-f018:**
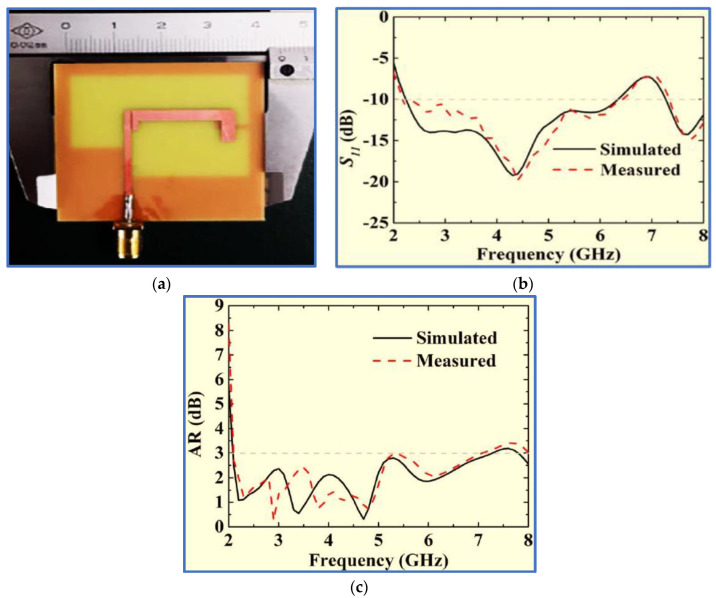
(**a**) Fabricated prototype of a microstrip feeding CP slot antenna. (**b**) Simulated and measured S11 results. (**c**) Simulated and measured AR results [18].

**Figure 19 micromachines-13-01048-f019:**
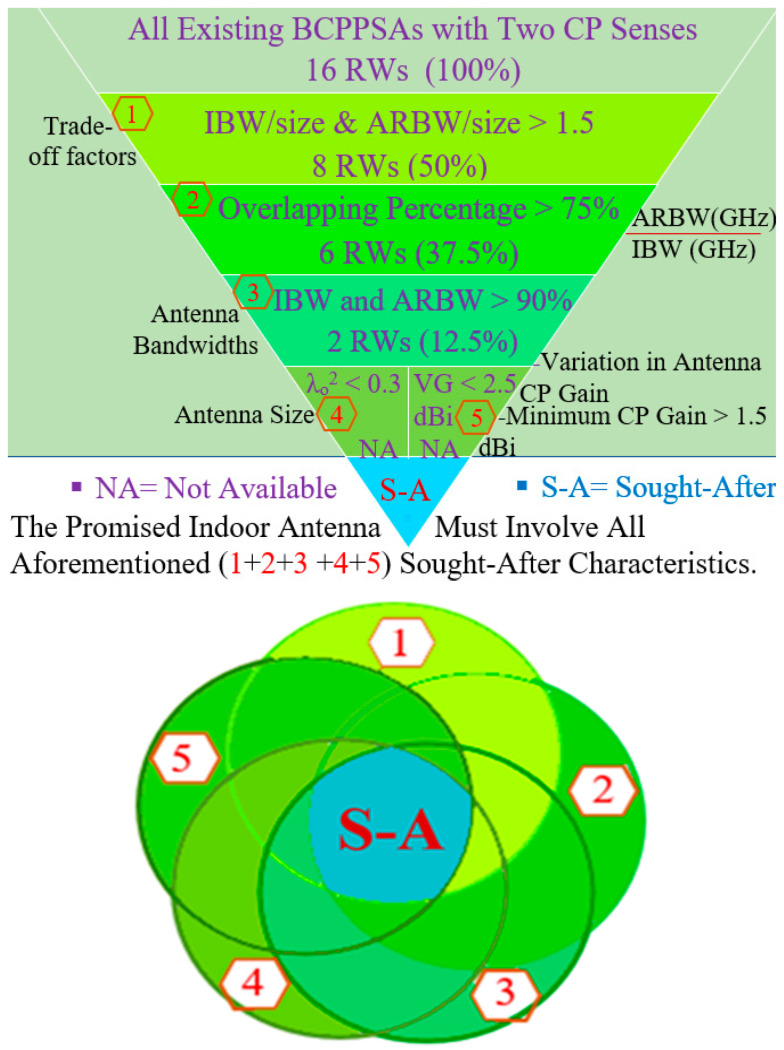
Analysis process of the BCPPSAs’ performances.

**Figure 20 micromachines-13-01048-f020:**
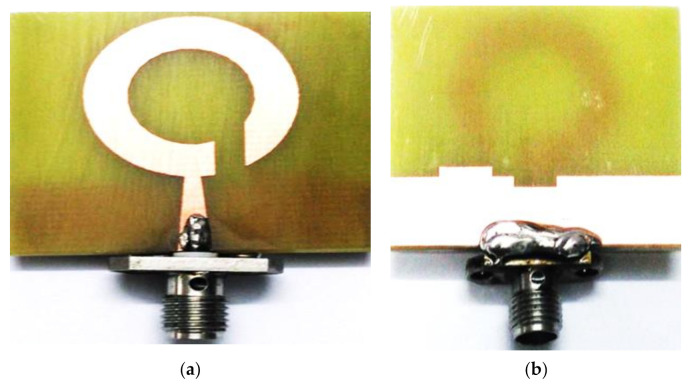
A microstrip-fed monopole: (**a**) front side; (**b**) back view; (**c**) measured radiation patterns at 5.2 GHz [117].

**Figure 21 micromachines-13-01048-f021:**
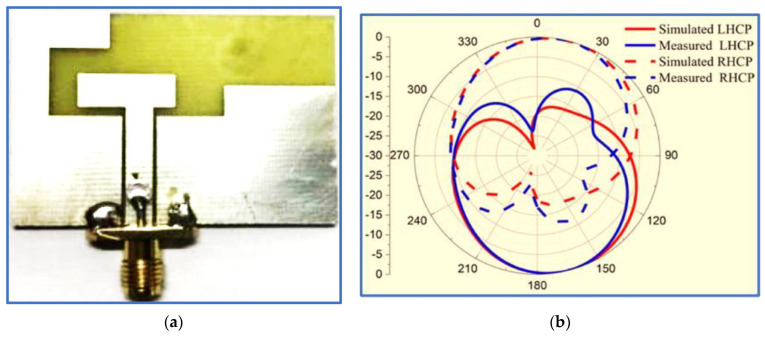
A CPW-fed monopole antenna. (**a**) The fabricated antenna. (**b**) The radiation patterns of the CP print monopole antenna at 4.8 GHz (xoz-plane) [16].

**Figure 22 micromachines-13-01048-f022:**
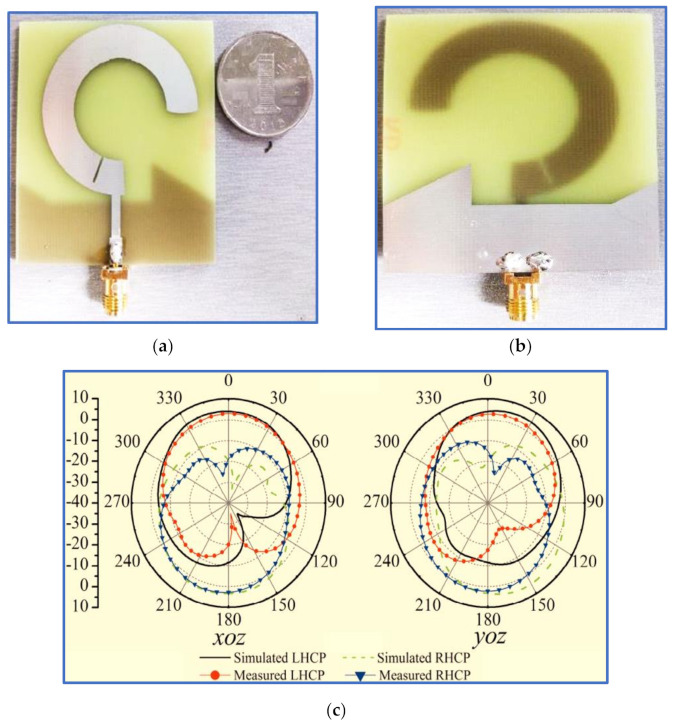
The geometry of a C-shaped monopole antenna: (**a**) front view; (**b**) back view; (**c**) radiation pattern at 2.4 GHz [123].

**Figure 23 micromachines-13-01048-f023:**
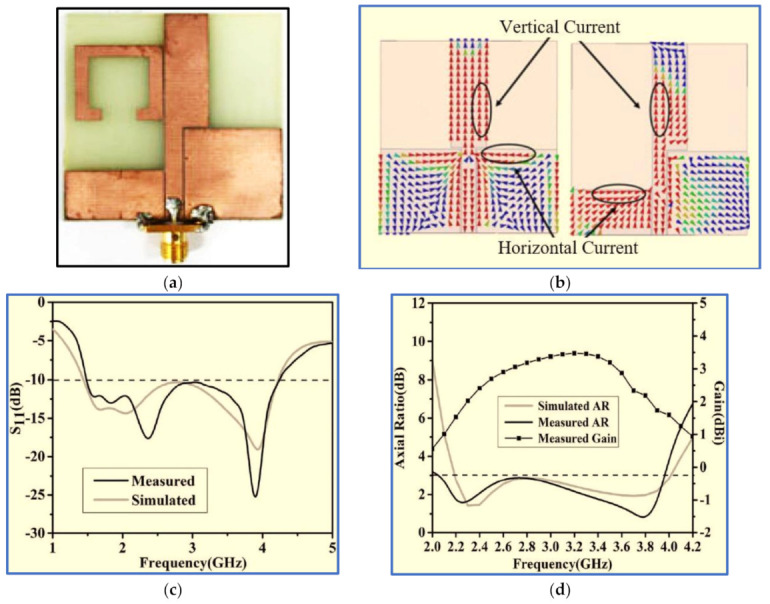
(**a**) A prototype of the CPW-fed rectangular monopole antenna. (**b**) The effect of the asymmetry ground plane on the surface current. (**c**) The S11 result. (**d**) The AR and gain results [127].

**Figure 24 micromachines-13-01048-f024:**
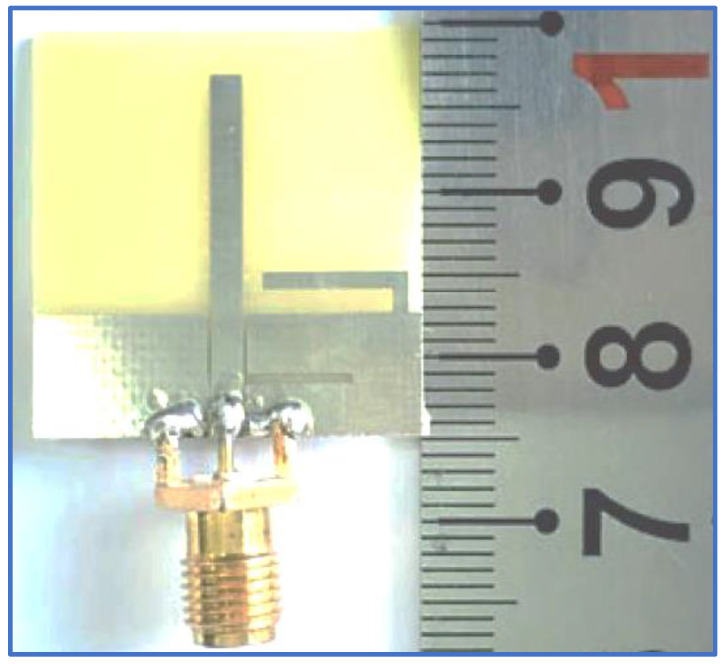
The geometry of a CPW-fed straight monopole antenna [128].

**Figure 25 micromachines-13-01048-f025:**
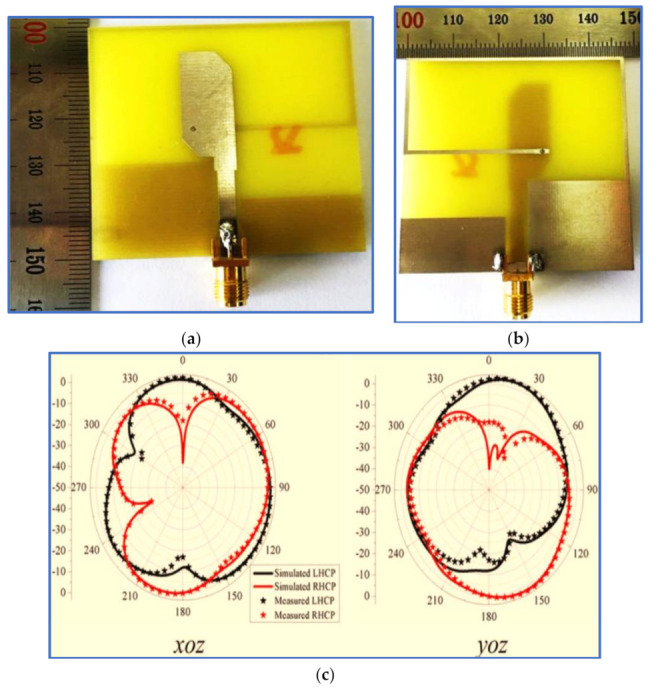
A modified rectangular monopole antenna: (**a**) front side; (**b**) back side; (**c**) normalized radiation pattern at 4.7 GHz [134].

**Figure 26 micromachines-13-01048-f026:**
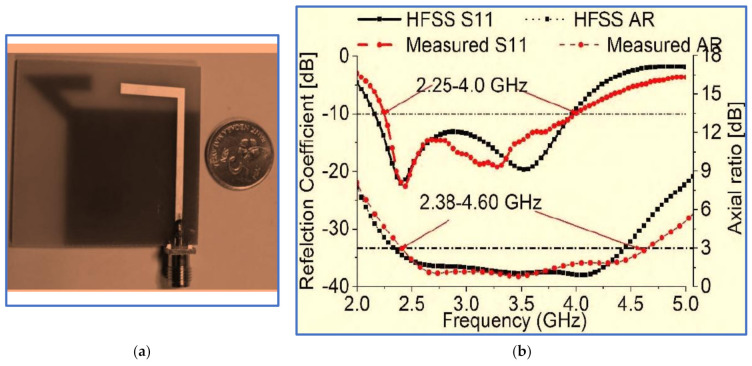
(**a**) A fabricated hook-shaped monopole antenna. (**b**) Comparison between the simulated and measured S11 and AR results [138].

**Figure 27 micromachines-13-01048-f027:**
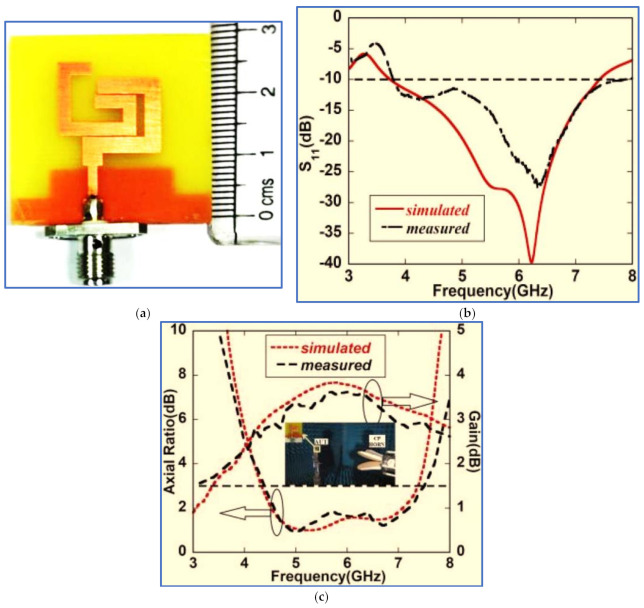
(**a**) A fabricated prototype of an inverted C-shaped monopole antenna; (**b**) S11 results; (**c**) AR and antenna gain results [139].

**Figure 28 micromachines-13-01048-f028:**
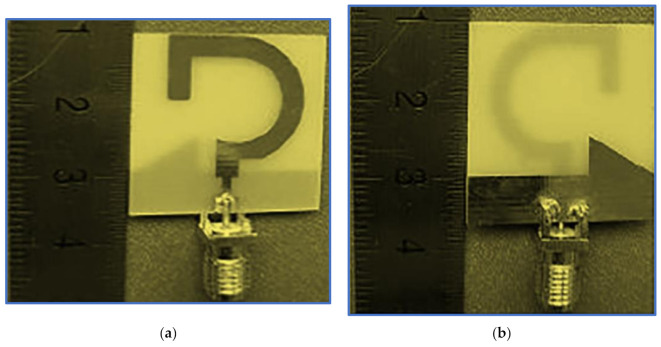
A prototype of the modified inverted C-shaped monopole antenna: (**a**) front view; (**b**) back view [141].

**Figure 29 micromachines-13-01048-f029:**
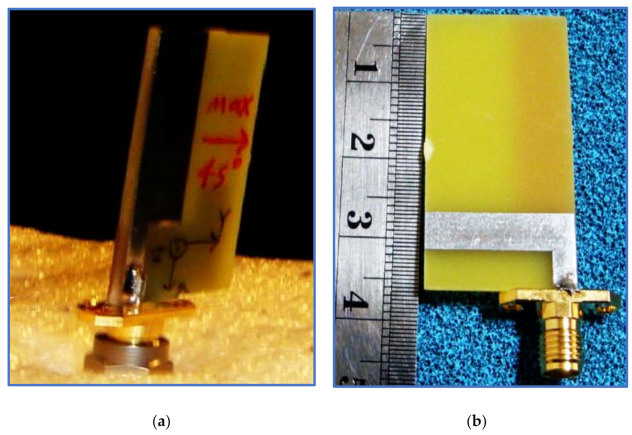
A prototype of the asymmetric fed rectangular monopole antenna: (**a**) side view; (**b**) back view [142].

**Figure 30 micromachines-13-01048-f030:**
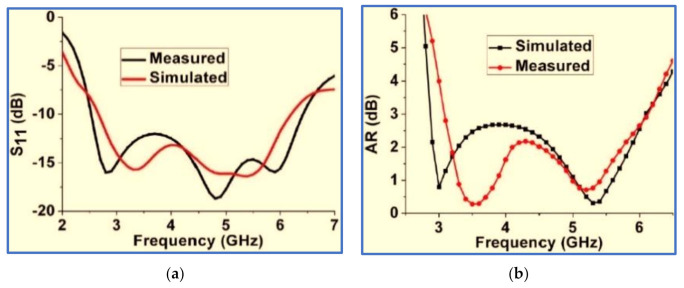
Comparison results for the asymmetric-fed rectangular monopole antenna: (**a**) S11 results; (**b**) AR results; (**c**) antenna CP gain [142].

**Figure 31 micromachines-13-01048-f031:**
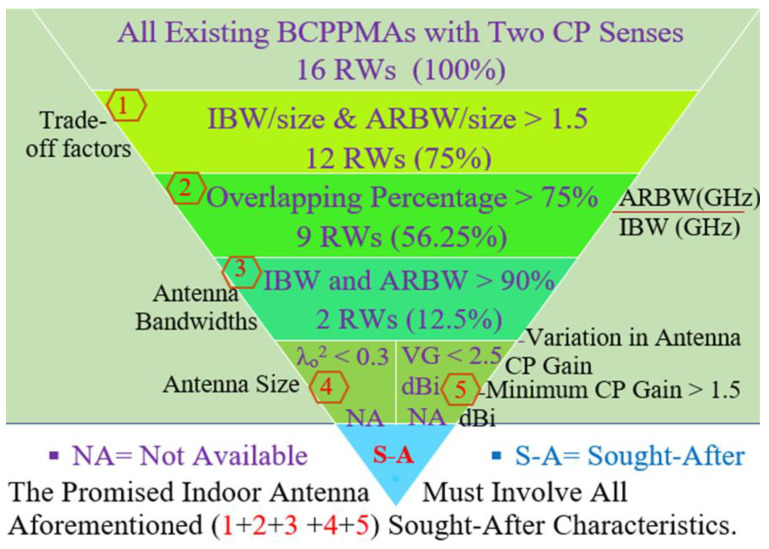
The analysis process of the BCPPMAs’ performances.

## Data Availability

Data is contained within the article.

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
