# Peer review of "Broadband Circular Polarised Printed Antennas for Indoor Wireless Communication Systems: A Comprehensive Review"

_micromachines, 2022, doi:10.3390/mi13071048_

Round 1
Reviewer 1 Report
The authors present Broadband Circular Polarised Printed Antennas for Indoor Wireless Communication Systems: A Comprehensive Review. The paper is well organized and equipped with the state of the artwork. The paper is good for acceptance, however, typos and grammatical mistakes need to be revised carefully.
Author Response
Dear Reviewer
Thank you for your comment. However, we have carefully checked the Manuscript's typos and grammar.
Our regards
Reviewer 2 Report
Authors are requested to cite the following paper:
Banerjee U, Karmakar A,
Saha A (2020). A review on circularly polarized
antennas, trends and advances.International
Journal of Microwave and Wireless Technologies
1–22. https://doi.org/10.1017/
S1759078720000331
Author Response
Dear Reviewer
Thank you for your reference suggestion. However, we have cited the given reference, which can be seen in ref [6]. (Line 75).
Our regards
[6] Banerjee, U.; Karmakar, A.; Saha, A. A review on circularly polarized antennas, trends and advances. J. Microw. Wirel. Technol. 2020, 12(9), 922-943.
Reviewer 3 Report
The authors must incorporate more research articles and also include the radiation pattern and its discussion.
Author Response
Dear Reviewer
Thank you for your comment and suggestion. However, as seen in the introduction, we have added more articles and references [3,6,12]. (Lines: 62, 75, 116]. However, the radiation patterns have been added to most figures (some articles don't have radiation pattern graphs).
The new radiation patterns graphs can be found as follows:
1- Figure 3 (lines 176 to 179).
2- Figure 4 (lines 221 to 223).
3- Figure 5 (lines 278).
4- Figure 6 (lines 282 to 286).
5- Figure 7 (lines 414)
6- Figure 9 (lines 458 to 459).
7- Figure 14 (lines 623 to 624).
8- Figure 15 (lines 684 to 685).
9- Figure 16 (lines 715 to 716).
10- Figure 17 (lines 752 to 753).
11- Figure 21 (lines 951 to 952).
12- Figure 22 (lines 1022 to 1023).
13- Figure 25 ( lines 1152 to 1153).
Our regards